# Electron confinement-enhanced green InP-based quantum dots for active-matrix LEDs displays

Ning Guo [1,2,3,6], Ke He[1,2,3,6], Hui Li [4 ✉], Tianchen Li[1,3], Fengmian Li[1], Jiangang Feng [4], Zhiyuan He [5 ✉], Lei Jiang [1,4] & Yuchen Wu [1,2,3 ✉]

The facet-selective growth of shells on green InP-based quantum dots result in their inferior electron confinement capabilities, posing a challenge for the realization of completely cadmium-free quantum dot light-emitting diode displays. Here, we develop a surface energy homogenization strategy based on ligand adsorption using n-octylamine and diphenylphosphine selenide, effectively suppressing selective growth of ZnSe on the InP (111) facet, resulting in strongly electron-confined InP/ZnSe/ZnS quantum dots with a quantum yield exceeding 92% and a full-width at half-maximum of 35 nm. The resulting quantum dot light-emitting diodes achieve a peak external quantum efficiency of 23.50% and a luminance exceeding $1.4 \times 10^5$ cd m$^{-2}$, with a 107.5-fold increase in device lifetime. Utilized asymmetric wettability-mediated assembly strategy, we achieved quantum dot arrays with an impressive resolution of 8460 PPI. Furthermore, integrating the quantum dots into an active-matrix LED display, we successfully demonstrate the display of both static and dynamic images.

Quantum dot light-emitting diodes (QLEDs) have emerged as a prominent technology for next-generation display and lighting applications, owing to their exceptional attributes such as low power consumption and high color purity[1-4]. For a prolonged period, the most advanced QLEDs have predominantly relied on cadmium (Cd)-based quantum dots (QDs) from the II-VI group[5-8]. However, the inherent toxicity of Cd has significantly hindered their large-scale commercial adoption. As a result, indium phosphide (InP)-based QDs have gained considerable attention as viable substitutes for Cd-based QDs. To date, the highest reported external quantum efficiencies (EQEs) for red and green InP-based QLEDs have reached 26.6% and 26.7%, respectively[9,10]. Despite this promising progress, the facet-selective growth of shell layers during the synthesis of InP-based QDs presents a substantial challenge for

further advancing InP-based QLEDs to compete with their Cd-based counterparts[11,12].

Recent studies have clearly demonstrated that the inferior performance of green InP-based QLEDs is primarily due to insufficient electron concentration, a key finding that deepens the understanding of green InP-based QLEDs[10]. In comparison to CdSe-based QDs, InP-based QDs exhibit a lower effective electron mass (0.07 $m_0$ versus 0.13 $m_0$, where $m_0$ denotes the free electron mass)[2], necessitating a higher degree of shell uniformity to enhance electron confinement[4,13]. ZnSe has emerged as the preferred shell material for epitaxial growth on InP cores, due to its favorable electronic structure (type I), comparable crystal structure (zinc blende), and matched lattice parameters, with lattice parameters of InP, CdSe, and ZnSe being 5.87 Å, 6.08 Å, and 5.67 Å, respectively[12,14,15]. These attributes facilitate epitaxial growth of

[1]Key Laboratory of Bio-inspired Materials and Interfacial Science, Technical Institute of Physics and Chemistry, Chinese Academy of Sciences, Beijing, P. R. China. [2]College of Chemistry, Jilin University, Changchun, P. R. China. [3]School of Future Technology, University of Chinese Academy of Sciences (UCAS), Beijing, P. R. China. [4]State Key Laboratory of Bioinspired Interfacial Materials Science, Suzhou Institute for Advanced Research, University of Science and Technology of China, Suzhou, Jiangsu, P. R. China. [5]School of Materials Science and Engineering, Beijing Institute of Technology, Beijing, P. R. China. [6]These authors contributed equally: Ning Guo, Ke He. ✉e-mail: lihui-t22031@ustc.edu.cn; hezy@bit.edu.cn; wuyuchen@iccas.ac.cn

ZnSe on InP cores. However, ZnSe tends to show facet-selective growth on InP cores, which leads to stronger electron delocalization that deviates from the spherical ZnSe shell model[16]. Existing studies suggest that adjusting the reaction temperature and stabilizing the (111) facet can mitigate this selective growth[11,12], but a deeper understanding of the mechanism underlying isotropic growth of ZnSe and its effects on QLED performance remains elusive.

Here, we develop a surface energy homogenization strategy that effectively passivates the highly reactive In$^{3+}$ on the InP (111) facet using a combination of n-octylamine and diphenylphosphine selenide (DPP-Se) ligands, leading to homogenization of the surface energies across all three facets and achieving uniform shell growth. The prepared InP/ZnSe/ZnS QDs exhibit strong electron confinement and demonstrate excellent optical performance, including a photoluminescence quantum yield (PLQY) exceeding 92% and a full-width at half-maximum (FWHM) of 35 nm. When integrated into QLEDs, they exhibit a peak EQE of 23.50% and a peak luminance exceeding $1.4 \times 10^5$ cd m$^{-2}$. The enhanced electron confinement of these QDs also mitigates electron leakage-induced damage to the hole transport layer (HTL), resulting in a 107.5-fold increase in device lifetime. Furthermore, through the asymmetric wettability-mediated assembly strategy[17–19], we fabricate high-resolution QLED devices and reduce the pixel size to 1.5 μm, achieving an impressive resolution of 8,460 pixels per inch (PPI). The high-resolution devices exhibited minimal performance degradation upon size reduction (from 20 μm to 1.5 μm), maintaining an average peak EQE exceeding 20% and exhibiting comparable performance to thin-film devices. Finally, by integrating the QDs into an active-matrix LED display, we successfully demonstrate the display of both static and dynamic images.

## Results

### Synthesize quantum dots with strong electron confinement

The facet-selective growth of ZnSe on InP cores is one of the factors impeding the synthesis of strongly electron-confined (SEC) InP-based QDs[2,11]. This facet-selective growth results in the formation of weakly electron-confined (WEC) InP-based QDs with irregular tetrapod shape (Fig. 1a, top). On one hand, the tetrapod QDs are unable to effectively confine electrons within the core for radiative recombination with holes, leading to electron delocalization and capture by surface traps, which induces severe non-radiative recombination and spectral broadening[11]. On the other hand, when WEC InP-based QDs are integrated into QLED devices, they result in significant charge leakage (Fig. 1b, top). The highly delocalized electrons in the InP-based QD layer recombine with holes in the HTL, not only causing pronounced parasitic emission peak in the electroluminescence (EL) spectrum[20,21], but also leading to the degradation of the HTL[21–23]. These issues severely constrain the EQE and the operational lifetime of the devices[24]. In contrast, the SEC InP-based QDs obtained from uniform shell growth not only effectively suppress electron delocalization (Fig. 1a, bottom), but also inhibiting charge leakage and thereby enhancing device performance (Fig. 1b, bottom)[20].

To delve into the mechanism of facet-selective growth of ZnSe on InP cores, surface energy simulations were conducted using density functional theory (DFT) (Supplementary Fig. 1). The calculated results indicate that the surface energies of the (111), (100) and (110) facets are 0.256 eV Å$^{-2}$, 0.187 eV Å$^{-2}$, and 0.068 eV Å$^{-2}$, respectively, with the (111) and (100) facets exhibiting significantly higher surface energies than the (110) facet. This discrepancy is attributed to the In-rich state on the (111) and (100) facets, where the (111) facet demonstrates greater instability and reactivity due to high density of surface dangling bonds[11,25]. In contrast, the (110) facet forms a self-passivated surface, with stoichiometric cations In$^{3+}$ and anions P$^{3-}$ rendering it the lowest energy[26,27]. To modulate the surface energy, the use of short-chain amine and organophosphine ligands was adopted as an effective strategy[28,29]. DFT calculations were employed to compare the

adsorption effects of three different ligand combinations: oleic acid (OA) with tri-n-octylphosphine (TOP)-Se, n-octylamine with TOP-Se, and n-octylamine with DPP-Se (Supplementary Fig. 2). The results revealed that the combination of n-octylamine with DPP-Se (Fig. 1c) can most effectively homogenize the surface energies of the (111), (100) and (110) facets, with specific values presented in Fig. 1d and Supplementary Table 1. Additionally, the formation energies of the three ligand combinations attached onto three facets were calculated (Fig. 1e and Supplementary Table 1). The results show that the formation energy of the n-octylamine with DPP-Se combination on the (111) facet (−38.98 eV) is significantly lower than that on the (100) facet (−14.48 eV) and (110) facet (−0.94 eV), and also lower than that of the other two ligand combinations. This suggests that the n-octylamine with DPP-Se combination is more likely to bind to the (111) facet, thereby facilitating the mitigation of the epitaxial growth rate of ZnSe on the (111) facet[30].

We synthesized InP cores according to a previously reported method with some modifications (Supplementary Notes 1 and 2). The InP cores exhibit a first exciton absorption peak at 445 nm with a valley-to-depth (V/D) ratio of 0.47 (Supplementary Figs. 3 and 4). Based on the computational results, we employed three different ligand combinations—OA with TOP-Se, n-octylamine with TOP-Se, and n-octylamine with DPP-Se—during the ZnSe growth process (Fig. 1f), followed by the growth of ZnS shells to obtain three types of InP/ZnSe/ZnS QDs (see details in "Methods"). Transmission electron microscopy (TEM) images reveal that the morphologies of the QDs transition from a tetrapod shape to an intermediate shape and finally to a spherical shape (Fig. 1g and Supplementary Fig. 5). Energy-dispersive spectroscopy element mapping demonstrates the distribution of Se in QDs synthesized with OA and TOP-Se closely aligns with the pod-like morphology, demonstrating more pronounced facet-selective growth as the ZnSe shell thickens (Supplementary Fig. 6)[31,32]. In contrast, the InP/ZnSe/ZnS QDs synthesized using n-octylamine with DPP-Se possess a highly symmetric core/multishell structure with uniform ZnSe shell growth (Supplementary Fig. 7). In addition, X-ray diffraction in Fig. 1h also shows that the (111) facet diffraction peak intensity of InP/ZnSe/ZnS QDs synthesized with a combination of n-octylamine and DPP-Se ligands is the lowest, which is consistent with the high-resolution TEM images analysis (Supplementary Fig. 8). These observations collectively indicate that our ligand combination successfully suppresses the selective growth of ZnSe on the (111) facet[31]. Fourier-transform infrared spectra and X-ray photoelectron spectroscopy (XPS) of N 1$s$ and In 3$d$ confirms that DPP and n-octylamine are successfully introduced and coordinated on the surface of the QDs (Supplementary Fig. 9).

### Optical performance characterization of quantum dots

During the synthesis process, the InP/ZnSe QDs synthesized using the combination of n-octylamine and DPP-Se exhibit higher PLQY (Supplementary Figs. 10 and 11a) and a significantly higher V/D value of 0.59 compared to 0.22 for that synthesized with OA and TOP-Se (Fig. 2a), indicating a more effective and uniform growth of ZnSe. Additionally, the InP/ZnSe/ZnS QDs synthesized with n-octylamine and DPP-Se did not exhibit significant spectral broadening during the ZnSe shell layers growth (Fig. 2b), ultimately exhibiting a smaller peak redshift (537 nm) and a narrower FWHM (35 nm), in comparison to the QDs synthesized with OA and TOP-Se (542 nm, FWHM = 42 nm) (Fig. 2c and Supplementary Fig. 11b). We employed transient absorption spectroscopy to analyze the exciton dynamics of two types of QDs (Fig. 2d). Both of the QDs exhibit two distinct bleaching peaks at ~510 nm and ~445 nm, which are attributed to the 1S$_e$-1S$_{3/2}$ and 1S$_e$-2S$_{3/2}$ transition, respectively[33,34]. The bleaching recovery dynamics of ground-state bleaching (GSB) peak (~510 nm) of two types of QDs were analyzed by fitting the decay curves with a tri-exponential function (Supplementary Table 2). The GSB of QDs synthesized with OA and TOP-Se shows

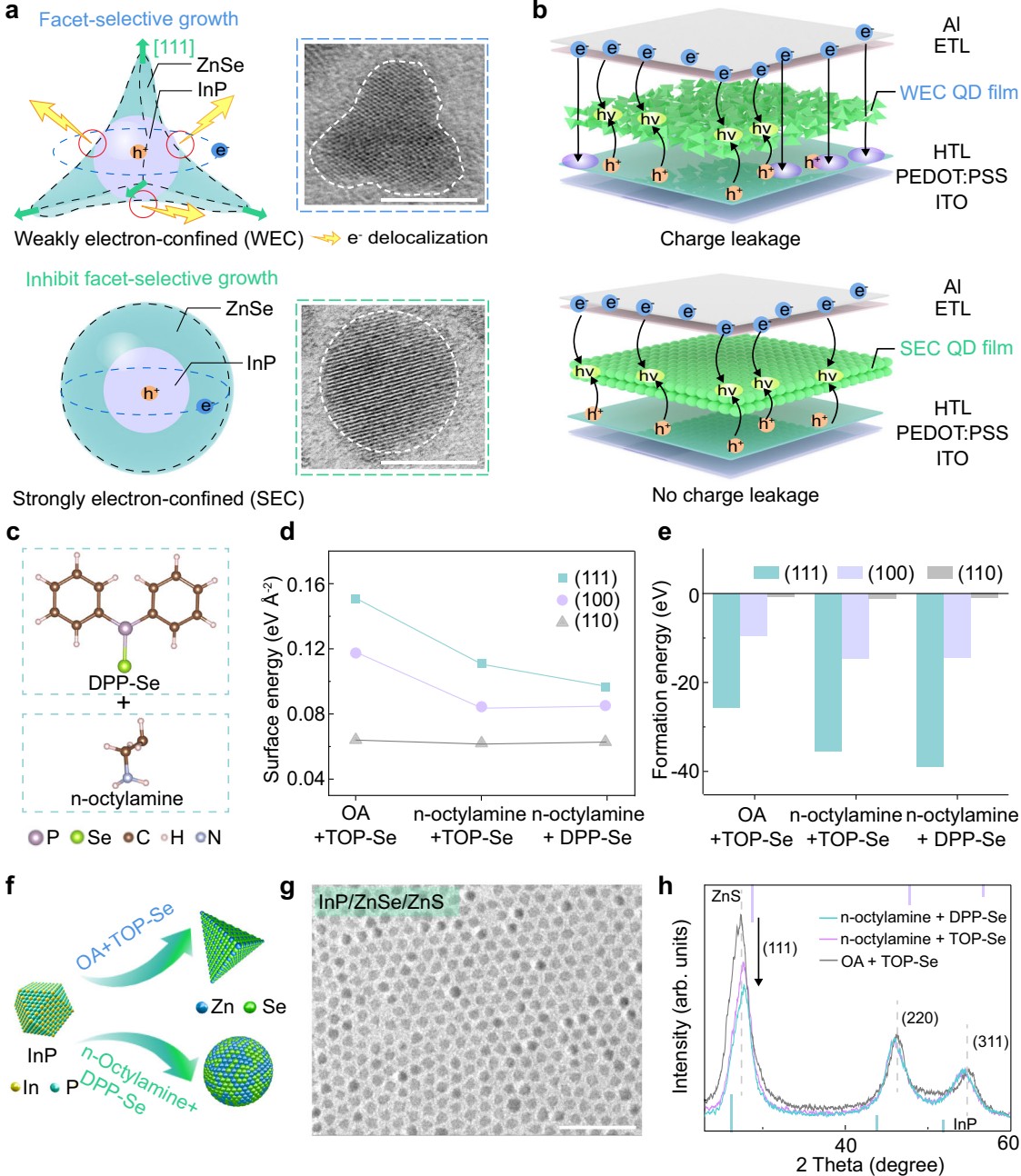

**Fig. 1 | Ligand-assisted surface energy homogenization strategy to synthesize SEC InP/ZnSe/ZnS QDs. a** Schematic illustration and representative high-resolution TEM images for two types of InP/ZnSe QDs. Scale bar, 5 nm. **b** Schematic illustration of charge leakage in QLEDs based on two types of QD films. **c** Molecular structures of ligand combination of n-octylamine and DPP-Se. **d** Calculated surface energies of (111), (100) and (110) facets of InP after adsorption of three different combinations of ligands (OA and TOP-Se, n-octylamine and TOP-Se, n-octylamine and DPP-Se). **e** Calculated formation energies of the three different combinations of ligands on the three facets. **f** Schematic illustration of the ZnSe growth with the two ligand combinations (OA and TOP-Se, n-octylamine and DPP-Se). **g** TEM image of SEC InP/ZnSe/ZnS QDs. Scale bar, 50 nm. **h** X-ray diffraction patterns of the QDs synthesized with the three different combinations of ligands.

pronounced bleaching and rapid decay dynamics, indicating poor defect passivation at the InP core and ZnSe interface[35]. In contrast, the GSB peak of QDs synthesized with n-octylamine and DPP-Se ligands demonstrates slower decay dynamics, suggesting effective exciton recombination[36,37]. These characteristics collectively demonstrate that the combination of n-octylamine and DPP-Se ligands during synthesis enables the uniform growth of the ZnSe shell, resulting in uniform size distribution and strong electron confinement.

The fluorescence decay lifetimes of the two types of QDs were obtained using time-resolved photoluminescence (TRPL) spectroscopy (Fig. 2e). Compared to the photoluminescence (PL) lifetime of WEC QD solution (57.95 ns), the SEC QD solution exhibited a longer PL lifetime of 84.83 ns, indicating effective passivation of defects and suppression of non-radiative recombination[38]. Additionally, the SEC QDs achieved a PLQY of 92.28%, significantly surpassing the 64.46% of the WEC QDs, and exhibited superior photostability under UV irradiation (Fig. 2f and Supplementary Table 3). This enhanced performance is attributed to the superior electron confinement, which effectively isolates electrons from surface traps and thereby significantly reducing non-radiative recombination via exciton-surface coupling[39,40]. Concurrently, this confinement promotes greater overlap of electron and hole wavefunctions, which enhances the exciton

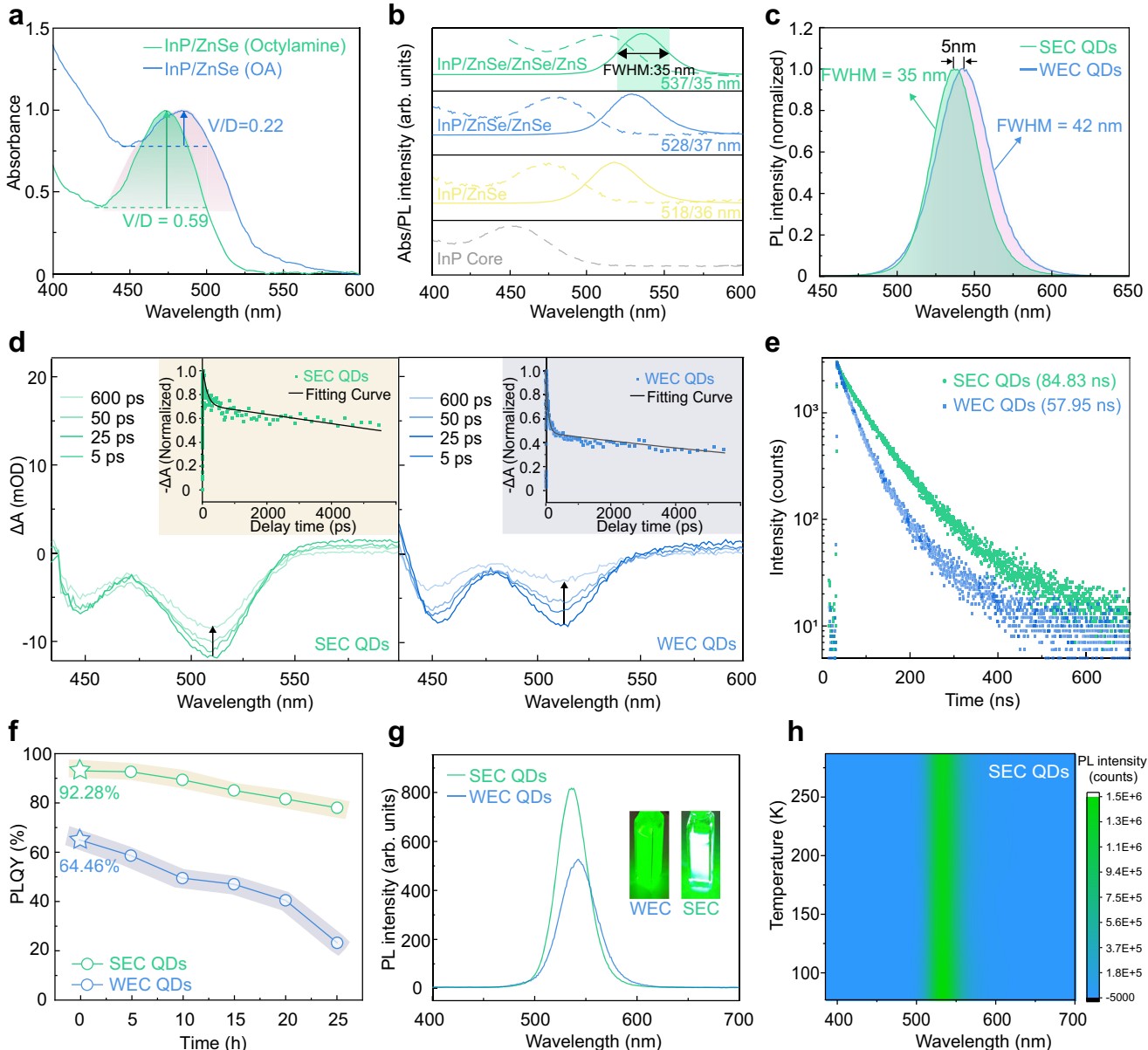

**Fig. 2 | Optical performance characteristics of SEC and WEC InP/ZnSe/ZnS QDs.**
**a** Absorption spectra and V/D values of InP/ZnSe QDs synthesized with two different combinations of ligands (OA and TOP-Se, n-octylamine and DPP-Se).
**b** Evolution of absorption spectra (dashed line) and PL spectra (solid line) during the synthesis process of SEC QDs. **c** Comparison of FWHM of PL spectra for two types of InP/ZnSe/ZnS QDs. **d** Transient absorption spectra and kinetic traces of the SEC (left) and WEC InP/ZnSe/ZnS QDs (right). **e** PL decay curves of two types of InP/ZnSe/ZnS QDs. **f** The decay curve of the PLQY for two types of InP/ZnSe/ZnS QDs under continuous ultraviolet light irradiation. **g** PL intensity at the same OD value of two types of InP/ZnSe/ZnS QDs (The inserts display photographs of two types of QD solutions taken under illumination at 365 nm). **h** Temperature-dependent PL spectra of SEC InP/ZnSe/ZnS QDs.

binding energy and boosts the radiative recombination rate. This dual mechanism—promoting radiative recombination while suppressing non-radiative decay—is further corroborated by the higher PL intensity of the SEC QDs at an equivalent optical density (Fig. 2g) and their exceptional stability in temperature-dependent PL measurements (Fig. 2h and Supplementary Fig. 12).

### Device performance characterizations
Next, we integrated the two types of QDs as emission layers into QLED devices (ITO/PEDOT:PSS/PF8Cz/QD/ZnMgO/Al) and demonstrated the enhanced EL performance of the QLED devices based on SEC QD films compared to those based on WEC QD films (see details in "Methods" and Supplementary Note 3). Cross-sectional TEM image of the QLED devices revealed the sequential stacking of the functional layers (Fig. 3a). Surface roughness of the two QD films was characterized using atomic force microscopy. Due to the irregular tetrapod shape of the WEC QDs, the QDs were randomly arranged within the film, resulting in a rough surface, with the atomic force microscopy image showing a surface roughness ($R_q$) of 4.37 nm (Supplementary Fig. 13a,b). In contrast, the SEC QD films exhibited a highly uniform and smooth surface (Supplementary Fig. 13c,d), with a smaller $R_q$ of only 0.64 nm. Additionally, the increased absorption intensity in the UV-visible absorption spectra also indicated a denser packing in the SEC QD films (Supplementary Fig. 14)[41].

The EL spectra under various voltages revealed that the EL emission peak of the QLED based on WEC QD films was located at 552 nm with a FWHM of 52 nm, whereas the EL emission peak of the QLED based on SEC QD films was located at 542 nm with a FWHM of 43 nm

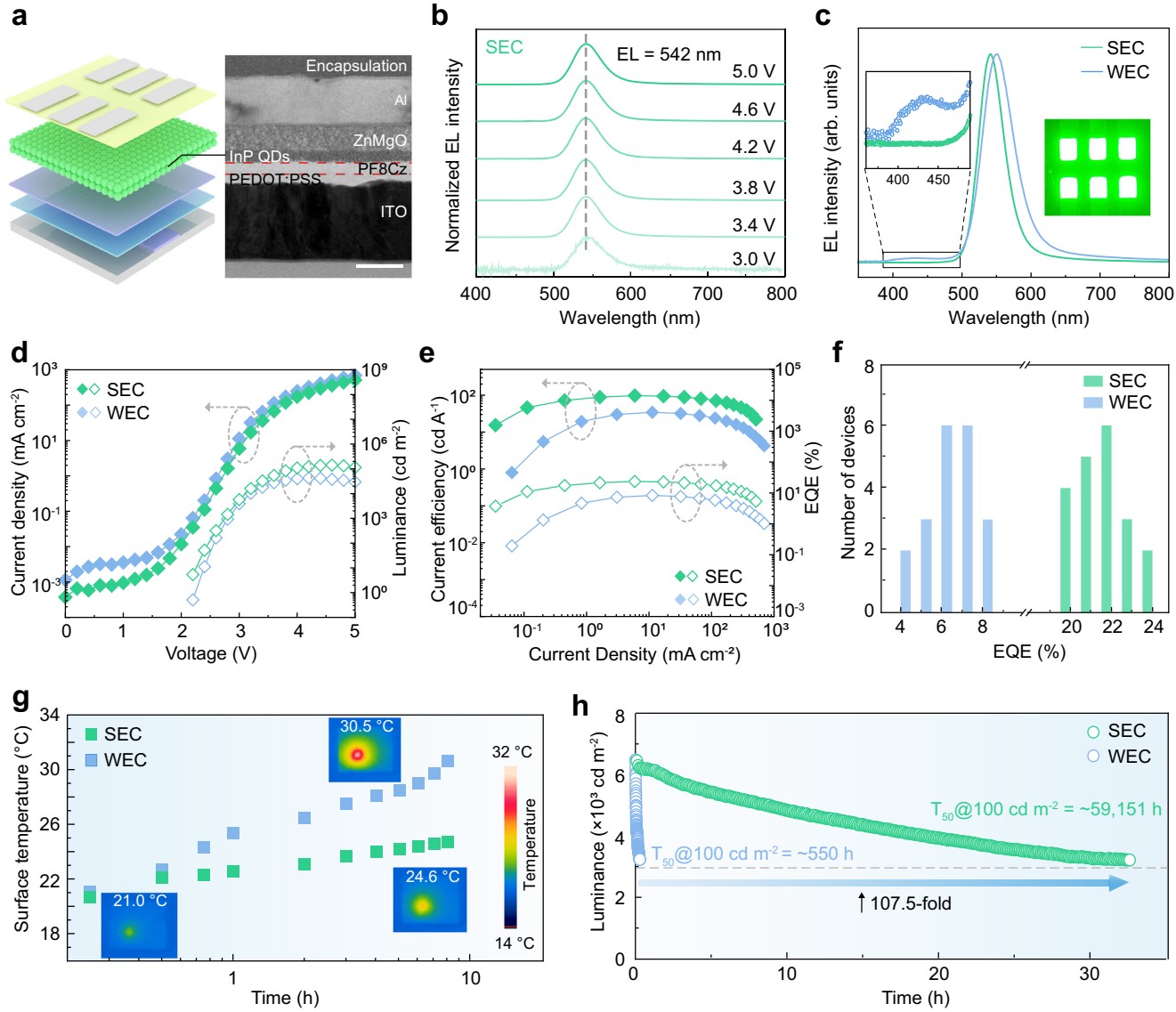

**Fig. 3 | The performance characterization of devices. a** Schematic diagram of the QLED device and corresponding cross-sectional TEM image. Scale bar, 100 nm. **b** EL spectra of device based on SEC QD films under different voltages. **c** The EL spectra of devices based on SEC QD films and WEC QD films at the same current density (Inset: photograph of an operational device based on SEC QD films). **d** Current density-voltage-luminance (*J-V-L*) characteristics of two types of QLEDs. **e** Current efficiency-current density-EQE characteristics of two types of QLEDs. **f** Histograms showing the peak EQEs of the two types of devices. **g** Time evolution of the surface temperature of two types of QLEDs after continuous operation at room temperature (20.0 °C). Data was collected using an infrared thermal imager with QLED devices operating under constant current density. **h** The decay curve of luminance over operating time for the two types of devices at an initial luminance of ~6500 cd m⁻².

(Fig. 3b). The corresponding CIE coordinates (0.320, 0.658) of the SEC QLED indicated a green emission with higher color purity than the WEC QLED, which is favorable for application in ideal green displays (Supplementary Fig. 15). The EL spectra of the SEC QDs exhibit a redshift of ~5 nm compared to the PL spectra, whereas the EL spectra of the WEC QDs shows a 10 nm redshift along with significant broadening (Supplementary Fig. 16). Furthermore, in electric-field-dependent PL measurements, the WEC QDs showed a higher sensitivity to the electric field and demonstrate more pronounced PL quenching and red shift with increasing voltage compared to the SEC QDs (Supplementary Fig. 17). These observations indicate a weaker Stark effect in the SEC QDs under an electric field, which can be attributed to strong electron confinement that suppresses the delocalization of electron wavefunctions toward the shell surface[2,42]. Due to the strong electron delocalization driven by the electric field, the EL spectrum of the QLED based on WEC QD films showed a parasitic emission peak (see Fig. 3c

and Supplementary Fig. 18a), which is consistent with the intrinsic EL spectrum of PF8Cz (Supplementary Fig. 18b). This is attributed to highly delocalized electron leakage from the InP QD layer to the HTL, where they recombine with holes in the HTL[21,43]. In contrast, the SEC QDs did not produce parasitic emission, effectively preventing charge leakage[20].

The current density-voltage-luminance (*J-V-L*) curves for the devices based on two types of QDs are shown in Fig. 3d. The QLED based on SEC QD films exhibited a peak luminance of 141,325 cd m⁻², which is 3.33 times higher than that of QLED based on WEC QD films (42,450 cd m⁻²). Additionally, the QLED based on SEC QD films exhibited a lower current density, resulting in a higher peak EQE and peak current efficiency (Fig. 3e), reaching 23.50% and 97.26 cd A⁻¹, respectively. This represented a 2.8-fold improvement compared to the QLED based on WEC QD films, which exhibited a peak EQE of 8.32% and a peak current efficiency of 34.43 cd A⁻¹. Space charge limited current

(SCLC) calculations indicate that the QLED based on SEC QD films exhibited higher hole mobility and lower electron mobility compared to the QLED based on WEC QD films (Supplementary Fig. 19a-d and Supplementary Notes 4 and 5), which is beneficial for balancing hole and electron injection. Furthermore, the trap state density of the QLED based on SEC QD films was reduced, as calculated from the trap filling limit voltage ($V_{TFL}$) (Supplementary Fig. 19e)[44,45]. Capacitance-voltage (C-V) characteristics showed that the QLED based on SEC QD films exhibited a smaller capacitance compared to the QLED based on WEC QD films, indicating a reduction in charge accumulation at the interface between QD layer and the HTL/ETL (Supplementary Fig. 19f)[46]. Figure 3f presented statistical data of the peak EQE for 20 QLED devices of each QD type, demonstrating good reproducibility of our devices.

Additionally, we measured the change in surface temperature of the devices over time under a constant current density, and the rise in surface temperature of the devices based on SEC QD films was significantly slower than that of the devices based on WEC QD films (Fig. 3g). For operational lifetime of QLEDs, as shown in Fig. 3h, the device based on SEC QD films exhibited a higher half-lifetime ($T_{50}$) of 32.5 h at an initial luminance of 6,470 cd m$^{-2}$. The lifetime at different luminance values was fitted using the empirical equation $L_0^n \times T_{50} = constant$ ($L_0$, initial luminance)[10]. The acceleration factors (n) were derived from the fitting of the data presented in Supplementary Fig. 20. The calculated $T_{50}$ at 100 cd m$^{-2}$ for QLEDs based on SEC QD films reached 59,151 h, representing a 107.5-fold enhancement compared to lifetime of QLEDs based on WEC QD films ($T_{50} \approx 0.288$ h at 6,501 cd m$^{-2}$, equivalent to $T_{50} \approx 550$ h at 100 cd m$^{-2}$) (Supplementary Fig. 21).

We attributed the substantially improved operational lifetime to strong electron confinement of SEC QDs, which suppressed electron leakage to HTL and mitigates HTL degradation. To confirm the impact of charge leakage on HTL degradation, we fabricated QLEDs with identical structures using two types of QDs and operated them at a constant current density of 2.0 mA cm$^{-2}$ for 20 h. The top layers were then removed to measure the absorption and photoluminescence spectra of PF8Cz before and after device operation (Supplementary Fig. 22a). The results reveal that after the same operational time, the absorption of PF8Cz in devices based on WEC QDs shows marked redshift and decrease in intensity, accompanied by a significantly enhancements in the lower-energy emissions (Supplementary Fig. 22b-d). The emergence of additional peak assigned to C = O at 288.1 eV in C 1s demonstrated that this degradation can be attributed to the electro-oxidation of fluorene units in PF8Cz to fluorenone group induced by leakage electrons (Supplementary Fig. 22e-g)[42,47]. In contrast, the HTL in devices based on SEC QDs shows no evident degradation, suggesting that enhanced electron confinement helps to mitigate HTL degradation, thereby improving the operational stability of the devices. To further rule out potential interference from different ligands, we synthesized InP/ZnSe/ZnS QDs featuring a thinner ZnSe shell (-1 nm) compared to that of the SEC QDs (-2 nm), while maintaining the same ligands of n-octylamine and DPP-Se. The corresponding devices exhibited a lifetime of only $T_{50} \approx 3,457$ h at 100 cd m$^{-2}$ (Supplementary Fig. 23), which mainly attributed to the insufficient electron confinement of the thinner ZnSe shell. The result further confirmed that strong electron confinement is the key factor underlying the significant improvement in device lifetime.

### Active-matrix light-emitting diodes displays

To investigate the application of these SEC QDs in displays, we utilized an asymmetric wettability-mediated assembly strategy to assemble the QDs into microstructure arrays and integrated them into QLED devices (Fig. 4a)[17–19]. Specifically, under the confinement of a micropillar template featuring asymmetric wettability (hydrophilic top and hydrophobic sidewall) (Supplementary Fig. 24 and Supplementary Note 6),

the liquid film was divided into individual capillary bridges on the top of the pillars. Accompanied by the solvent evaporation and directional dewetting, QDs deposited uniformly and assembled into microstructure arrays. The assembly process is detailed in Supplementary Fig. 25. By manipulating the concentration of the QD solution, we achieved high-quality QD arrays (Supplementary Fig. 26). Fluorescence microscopy and scanning electron microscopy images revealed that the QD microarrays exhibited uniform size and luminance (Supplementary Fig. 27). By adjusting the micropillar dimensions of the template (Supplementary Fig. 28), we successfully reduced the pixel size from 20 μm to 1.5 μm, achieving a maximum resolution of 8,460 PPI (Fig. 4b). Remarkably, statistical analysis demonstrated that our high-resolution devices maintained an average EQE above 20% across the entire pixel size range from 20 μm to 1.5 μm (Fig. 4c,d). The maximum luminance and operational lifetime of high-resolution devices were also comparable to those of thin-film devices (Supplementary Fig. 29). This maintained high performance with decreasing pixel size is attributed to controllable solvent evaporation and directed dewetting in our assembly strategy, which effectively suppresses the pronounced coffee-ring effect in spin-coating (Supplementary Fig. 30)[9,19]. The uniform spherical morphology of SEC QDs and their well-ordered self-assembly enabled by this controlled assembly process collectively enables uniform EL across the high-resolution display panel (Supplementary Fig. 31). Subsequently, we successfully realized static display of high-resolution patterns (Supplementary Figs. 32 and 33).

We further integrate SEC QDs into a 1.85-inch active-matrix LED display with a resolution of 352 × 430, where each pixel can be independently controlled (the structure of the display panel is shown in Supplementary Fig. 34). Under the regulation of the driving current from the thin-film transistor array driving circuit, individual pixels can achieve continuous and precise grayscale tuning. Figure 4e successfully demonstrates a static image of a lizard. The enlarged fine image of the local structure of the lizard under microscope exhibits uniform luminescence between adjacent pixels at the micro level. Figure 4f and Supplementary Video 1 further presented a vivid video showing water drops flowing. The fabrication of this InP QD-based active-matrix LEDs displays paves the way for the widespread application of heavy-metal-free QDs in the field of display technology.

## Discussion

We report a surface energy homogenization strategy which can suppress the facet-selective growth of ZnSe on the (111) facet of InP, thereby obtaining InP/ZnSe/ZnS QDs with strong electron confinement. Their strong electron confinement capability significantly reduces charge leakage in the QD layer, thereby enhancing the peak EQE and peak luminance of QLEDs to 23.50% and 141,325 cd m$^{-2}$, respectively, and markedly increasing the operational lifetime of the devices by 107.5-fold. Utilized the asymmetric wettability-mediated assembly strategy, we achieve QD arrays with an impressive resolution of 8,460 PPI. By integrating the arrays into an active-matrix LEDs display, we successfully demonstrate dynamic display. This study provides an idea for commercializing heavy-metal-free QD display technology.

## Methods

Additional details regarding the materials and methods can be found in Supplementary Information ('Experimental section').

### Materials

Zinc acetate (Zn(OAc)$_2$, 99.99%), sulfur (S, 99.998%, powder), oleic acid (OA, 90%) chlorobenzene, acetone (98%) and isopropyl alcohol (98%) were purchased from Sigma-Aldrich. n-Octylamine (98%) was purchased from Macklin. Selenium (Se, 99.999%, powder), 1-octadecene (ODE, 90%) were purchased from Thermo scientific. Trioctylphosphine (TOP, 90%), diphenylphosphine (DPP, ≥ 95%),

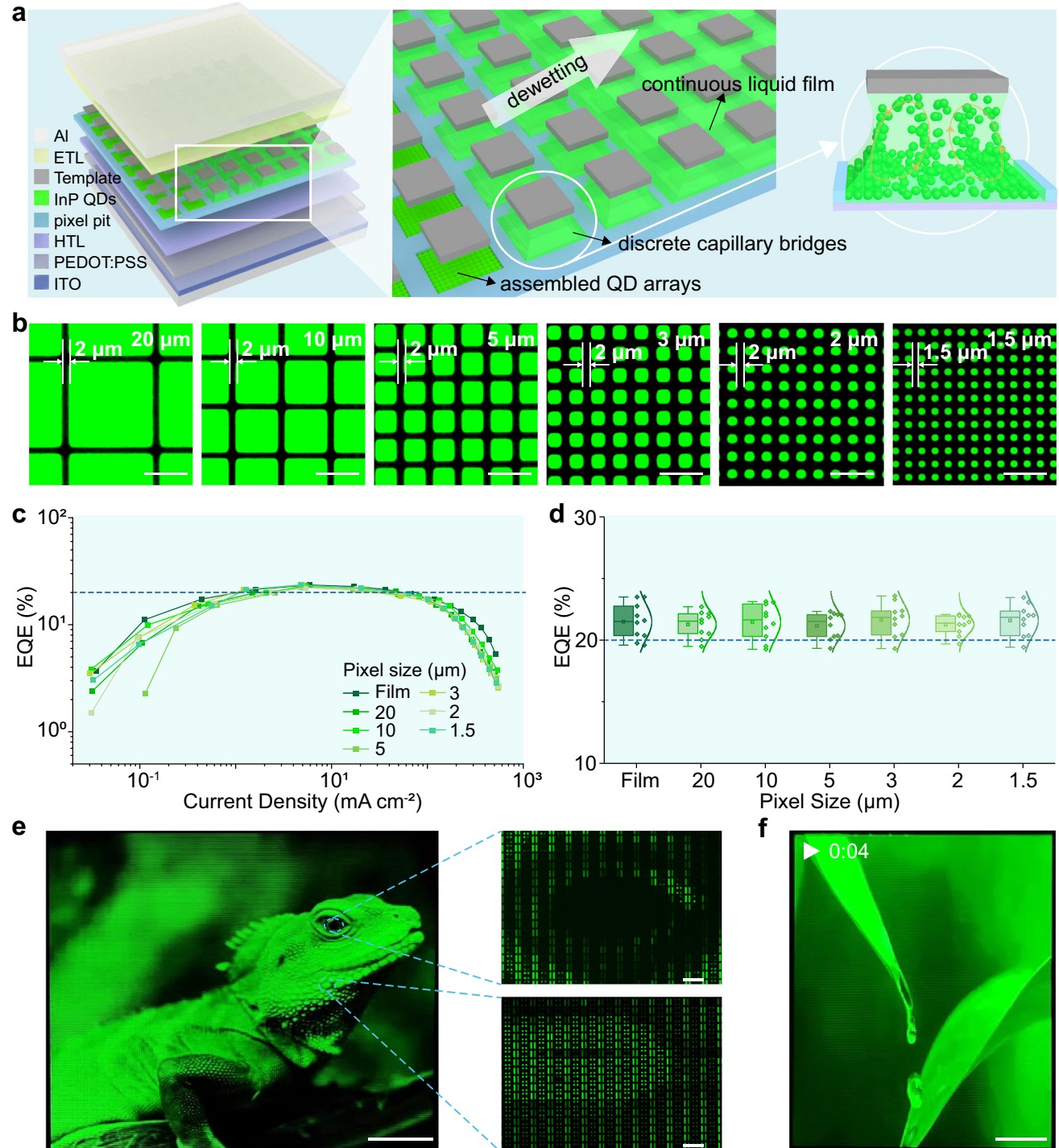

**Fig. 4 | Active-matrix LEDs displays based on SEC QDs. a** Schematic illustration of the assembly process for the SEC QD microstructure array. **b** Fluorescence microscopy images of high-resolution pixeled QD arrays. The pixel sizes are indicated in the upper right corner. Scale bar, 10 µm. **c** EQE-current density curves of QLED devices with different pixel sizes. **d** Statistics of maximum EQEs of high-resolution QLED devices with characteristic pixel size. The solid diamonds correspond to the EQE data of 10 devices with characteristic pixel size. The curves are the Gaussian fitting to the EQE distributions. **e** A lizard image displayed using the active-matrix LEDs displays (left) (scale bar, 10 mm) and its arrays of emissive pixel units of corresponding area after magnification by microscope (right) (scale bar, 200 µm). The photo is from a third-party website and has been authorized for use. **f** A video screenshot of the active-matrix LEDs displays (Supplementary Video 1). Scale bar, 10 mm.

hydrogen fluoride-pyridine (HF, 70 wt%), n-octane (99%) and n-hexanes (98%) were purchased from Aladdin. Ethanol (99.7%) and toluene (≥ 99.5%) were purchased from Sinopharm Chemical Reagent Co., Ltd. Poly((9,9-dioctylfluorenyl-2,7-diyl)-alt-(9-(2-ethylhexyl)-carbazole-3,6-diyl)) (PF8Cz) was purchased from Volt-Amp Optoelectronics Tech. Co., Ltd, Dongguan, China. Poly(3,4-ethylenedioxythiophene): poly (styrenesulfonate) (PEDOT:PSS, AI 4083) was purchased from Heraeus Deutschland GmbH & Co. KG.

## Synthesis of strongly electron-confined quantum dots

The shell growth was conducted using InP cores synthesized according to the procedure in Note 2 of Supporting Information. For the growth of shell, 1 mmol of $Zn(OA)_2$ and 10 mL of ODE were loaded in a 100 mL flask and purged with nitrogen and maintained at 120 °C for 30 min. Then 10 mL of n-octylamine was added into the system. The temperature was heated up to 160 °C with the as-prepared InP core in 2 ml toluene injected dropwise. Immediately, 0.12 mL of 7 wt% HF solution (see Supplementary Note 1 for the preparation process) was added and maintained the reaction for 5 min for removal of surface oxide. Then, the reaction solution was heated to 220 °C to grow the ZnSe layers. 2 mL of 0.4 M DPP-Se and 2 mL of 0.4 M $Zn(OA)_2$ were injected dropwise at 220 °C and kept the reaction for 1 h. Subsequently, the reaction solution was heated to 240 °C and 2 mL of 0.4 M DPP-Se and 2 mL of 0.4 M $Zn(OA)_2$ were inject dropwise for 1 h. Following the completion of this step, injected 2 mL of 0.4 M $Zn(OA)_2$ and 0.5 mL of 1.0 M TOP-S into the flask and reacted for 1 h when the solution was heated to 300 °C for the growth of ZnS layers. The resulting solution was then cooled to room temperature, washed with ethanol and n-hexane three times to remove impurities, and finally dispersed in n-octane for further characterizations.

## Synthesis of weakly electron-confined quantum dots

For the growth of shell, 1 mmol of $Zn(OA)_2$, 10 mL of ODE and 10 mL of OA were loaded in a 100 mL flask and purged with nitrogen and maintained at 120 °C for 30 min. Then the temperature of the flask was heated up to 160 °C before the as-prepared InP Core in 2 ml toluene was injected dropwise. Immediately, 0.12 mL of 7 wt% HF solution was added and maintained the reaction for 5 min for removal of surface oxide. Then, the reaction solution was heated to 220 °C to grow the ZnSe layers. 2 mL of 0.4 M TOP-Se and 2 mL of 0.4 M Zn(OA)2 were injected dropwise at 220 °C and kept the reaction for 1 h. Subsequently, the reaction solution was heated to 240 °C and 2 mL of 0.4 M TOP-Se and 2 mL of 0.4 M Zn(OA)2 were inject dropwise for 1 h. Following the completion of this step, injected 2 mL of 0.4 M Zn(OA)2 and 0.5 mL of 1.0 M TOP-S into the flask and reacted for 1 h when the solution was heated to 300 °C for the growth of ZnS. The resulting solution was then cooled to room temperature, washed with ethanol and n-hexane for 3 times to remove impurities, and finally dispersed in n-octane for further characterizations.

## Fabrication of devices

The indium tin oxide (ITO)-coated glass substrates were ultrasonically cleaned with detergent, deionized water, acetone, and isopropyl alcohol for 15 min each, and were then treated with $O_3$-plasma for 15 min. Subsequently, a hole injection layer (HIL) of PEDOT:PSS was spin-coated onto ITO substrates at 4000 rpm for 40 s, followed by annealing at 150 °C for 30 min. Then, these substrates were transferred into a nitrogen-filled glove box for spin-coating PF8Cz, InP QDs, and ZnMgO layers. PF8Cz was prepared by spin-coating 30 µL of 8 mg mL$^{-1}$ solution in chlorobenzene at 3000 rpm for 40 s, and then baked at 120 °C for 30 min. The QDs layer was prepared by spin-coating 30 µL of 20 mg mL$^{-1}$ QD solution at 2000 rpm for 30 s and baked at 80 °C for 7 min. The ZnMgO nanoparticles solution synthesized with the method in Supplementary Note 3 was spun-coating at 2000 rpm for 30 s and baked at 60 °C for 30 min. Al electrode (with a thickness of 100 nm) was deposited by thermal evaporation under a degree of vacuum of ≈ $2.5 \times 10^{-4}$ Pa. Ultimately, the devices were encapsulated using UV-curable epoxy resin and cover glass in a glove box.

The peel-off experiments were carried out on unencapsulated QLEDs in a nitrogen-filled glovebox to expose the PF8Cz layer. The Al cathode was first detached using polyimide tape. Then, the ZnMgO layer was washed away by spin-coating 30 µL of an acetic acid-ethanol solution (0.8% by volume) at 2000 rpm for 30 seconds, followed by the same process using 30 µL of pure ethanol. The QD layer was removed by spin-coating 30 µL n-octane at 2000 rpm for 30 seconds.

## Fabrication of high-resolution patterned devices

The ITO-coated glass substrates were ultrasonically cleaned with detergent, deionized water, acetone, and isopropyl alcohol for 15 minutes each, and were then treated with $O_3$-plasma for 15 min. Subsequently, PEDOT:PSS was spin-coated onto ITO substrates at 4000 rpm for 40 s, followed by annealing at 150 °C for 30 min. Then, these substrates were transferred into a nitrogen-filled glove box for spin-coating. The PF8Cz layer was spin-coated at 3000 rpm for 30 s using an 8 mg mL$^{-1}$ solution in chlorobenzene, followed by baking at 120 °C for 30 min. Next, fabricate photoresist pit arrays on these substrates by photolithography. Negative photoresist was spun-coated at 3000 rpm for 30 s and baked at 90 °C for 1 min. A custom mask was used to expose the photoresist on the photolithography machine for 5 s, followed by development, rinsing, and drying to create pixel pits of specific size. Then substrates with photoresist pits, 6 µL of 50 g mL$^{-1}$ QDs toluene solution and a micropillar template were integrated together (aligning each pit with each pillar on the template using the alignment system). A self-made pressure device was used to maintain proper and balanced pressure. The solution was evaporated at room temperature for 6 h to form a QDs array pattern on the substrate. Subsequently, The ZnMgO nanoparticles solution synthesized with the method in note S3 was spun-coating at 2000 rpm for 30 s and baked at 60 °C for 30 min. Al electrode was deposited by thermal evaporation under a based vacuum of ≈ $2.5 \times 10^{-4}$ Pa. Ultimately, the devices were encapsulated using UV-curable epoxy resin and cover glass in a glove box.

## Characterization of materials

TEM (JEOL, 2100 F, Japan) and TEM (JEOL, 7700 F, Japan) were used to evaluate the size and morphology of the QDs at 200 kV accelerating voltage. UV-vis absorption and PL spectra were measured by an Ocean Optics spectrophotometer (model PC2000-ISA). All scanning electron microscopy images were tested at 10 kV and 10 µA current using the Hitachi SU8010 (Japan) instrument. The surface morphology and roughness of the QD array were measured using atomic force microscope (Bruker Nano, ICON2-SYS). The QD lattice was acquired by in-situ double spherical aberration correction transmission electron microscope (JEM-ARM300F, Japan) at an operating voltage of 300 kV. Fourier-transform infrared spectroscope (Agilent, Excalibur 3100, USA) was used to collect covalent bond vibration signals. The Edinburgh instrument FLS 1000 was used to analyze the emission spectra, PLQY, and time-resolved PL spectra of QDs. All fluorescence images were captured using a microscope (Olympus DP80). X-ray diffraction data were acquired using an X-ray diffractometer (Bruker Nano Inc.) with monochromatized Cu $K_\alpha$ radiation ($\lambda$ = 1.5406 Å). XPS spectra were measured using an ESCALAB 250Xi system. Capacitance-voltage characteristics curves were performed using a semiconductor analyzer (LakeShore 8425, 4200A-SCS, B1500) with a modulating frequency of 30 KHz. The luminescence intensity distribution of the QD array was obtained by confocal imaging microscopy (ARsiMP-LSM). Transient absorption measurements were performed using femtosecond laser pulses from an 800 nm laser beam (1 kHz pulse repetition rate, 25 fs pulse duration) generated by a regenerative amplified Ti: Sapphire laser system (Coherent Co.). The output of the amplifier was split into two streams of pulses with a beam splitter. Residual stream was directed into an ultrafast spectroscopic system (Helios pump-probe system (Ultrafast Systems) to generate the white light continuum probe beam (300-2600 nm window with various optical filters). The pump wavelength was set at 370 nm. Delaying the probe pulse relative to the pump allowed a time window of up to 5 ns. The instrument response function was determined to be ≈ 200 fs using a cross-correlation procedure. The sample was filled in a 1 mm thick quartz cell for transient absorption experiment which was continuously swayed to prevent the optical damage.

## Characterization of devices

All devices were tested in a nitrogen-filled glove box using homemade test sockets. The *J-V-L* curves, EL spectra, current efficiency and EQE were measured using a commercial system (XPQY-EQE-Adv from Xipu Optoelectronics Technology Co., Ltd., Guangzhou). For lifetime testing, the encapsulated samples were measured in a nitrogen environment.

## DFT calculations

All calculations were carried out using the DFT with the generalized Perdew–Burke–Ernzerhof (PBE), and the projector augmented-wave (PAW) pseudopotential plane-wave method was implemented in the VASP code[48–52]. A $2 \times 2 \times 1$ Monkhorst-Pack k-point grid was used for all unitcell geometry optimization calculations. The plane-wave energy cutoff is set to 500 eV. The convergence criterion for the total energy tolerance per atom is $1 \times 10^{-6}$ eV atom$^{-1}$, and the maximum force tolerance is 0.02 eV Å$^{-1}$.

## Data availability

The data that supports the findings of this study are available from the corresponding author upon request. Source Data are provided with this paper. The raw data are available via Zenodo at https://doi.org/10.5281/zenodo.18232241. Source data are provided with this paper.

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

## Acknowledgements

J.G.F. acknowledges financial support from the National Key Research and Development Program of China (no.2024YFB3612800). Y.C.W. acknowledges funding support from the National Natural Science Foundation of China (nos. T2425026, 52173190 and 21988102) and Youth Innovation Promotion Association CAS (no. 2018034). H.L. acknowledges financial support from the Jiangsu Funding Program for Excellent Postdoctoral Talent (2025ZB663), the Basic Research Program of Jiangsu (BK20250477) and the China Postdoctoral Science Foundation under Grant Number 2025M781035.

## Author contributions

N.G., K.H., H.L. and T.L. fabricated the devices and collected the performance data of the QLEDs. N.G., K.H. and H.L. synthesized the materials. N.G., K.H. and H.L. wrote the manuscript. N.G., K.H., H.L., T.L., F.L. and J.F. conducted data analysis. Y.C.W. provided financial support. H.L., Z.H., L.J. and Y.C.W. directed the project. All authors contributed to the scientific discussion and modified the manuscript.

## Competing interests

The authors declare no competing interests.
