## [Transparent Peer Review file · Nature Communications]

Electron confinement-enhanced green InP-based quantum dots for active-matrix LEDs displays

Corresponding Author: Professor Yuchen Wu

Version 0:

Reviewer comments:

Reviewer #1

(Remarks to the Author)

This manuscript presents the synthesis of InP quantum dots (QDs) and their application in QD-light emitting diode (QLED) and active matrix LED display. From the perspective of InP QDs synthesis, the authors utilized different combinations of ligands such as OA:TOP-Se, octylamine:TOP-Se, and octylamine:DPP-Se for the ZnSe shell formation to mitigate its facet-selective growth and promote the homogeneity. The final InP/ZnSe/ZnS QDs exhibited a quasi-spherical shape by using octylamine:DPP-Se combination. They implemented these InP QDs for the fabrication of QLED and an active matrix green display. The QLED exhibited a peak external quantum efficiency (EQE) of 23.50% with a life span of 59,151 h at 100 cd m⁻². However, the findings of this work are not sufficiently novel, making it not suitable for consideration in Nature Communications; therefore, I do not recommend it for publication. The detailed concerns are discussed below:

1- The authors demonstrated that using octylamine:DPP-Se, they eliminated the facet-selective growth; however, there is a rich literature available where octylamines in combination with TOP-Se have shown the growth of homogenous ZnSe shells on InP QDs, affecting the novelty of this work. For example consider the representative works below:

[1] Nature 575, 634–638 (2019)

[2] Nature 635, 854–859 (2024)

2- In the XRD analysis (Fig. 1h), the authors claimed that the reduced (111) peak intensity in SEC QDs shows suppressed ZnSe growth on the (111) facet. However, the XRD peak intensities are also influenced by multiple factors, including crystal orientation, size effects, etc.

3- While Figures 2f-h demonstrate superior photostability of SEC QDs through PLQY decay under UV irradiation, solution-phase PL intensity, and temperature-dependent PL spectra, the authors provided no mechanistic explanation linking these observations to enhanced electron confinement.

4- Regarding the device performance, the authors attributed 107.5-fold operational lifetime enhancement in SEC QD device exclusively to suppressed electron leakage into the HTL via strong electron confinement lacks mechanistic validation. There could be other parameters involved, such as different ligands that could passivate the QDs' surface and affect their overall stability under electric field.

5- The authors provided no experimental evidence demonstrating how electron leakage initiates HTL degradation, nor did they consider factors like interfacial defect generation across different layers.

6- The attribution of the 430 nm parasitic peak in WEC device solely to electron leakage-induced recombination in the HTL requires further investigations. To strengthen this claim the EL spectra from PF8Cz-based control device will be helpful to verify whether its intrinsic emission aligns with the 430 nm feature.

7- The maximum EQE of the QLED reported is 23.50% which is lower than the recent reported EQE (26.68%) for the green InP QLED, why? (Nature 635, 854–859 (2024))

8- The FWHM of the EL spectrum increased to 43 nm, much broader than its PL spectrum, which is 35 nm. No attention is paid to this factor, like what caused such broadening, as it affects the color purity.

9- The T50 at 100 cd m⁻² for QLED based on SEC InP QDs reached 59,151 h only, whereas the recent best reported case for green InP QLED demonstrated a T95 of 89,900 h under similar brightness conditions.(Nature 635, 854–859 (2024))

Reviewer #2

(Remarks to the Author)

The authors report that the combined use of n-octylamine and DPP-Se enables isotropic ZnSe shell growth on InP cores, thereby enhancing electron confinement. They achieve a QY above 92%, an FWHM as narrow as 35 nm, a QLED EQE exceeding 23.5%, and a maximum brightness above 140,000 cd m⁻². They also demonstrate an asymmetric wettability-mediated assembly strategy for QD arrays with a resolution of 8,460 ppi, integrated into an AM-display prototype. It is known that ZnSe tends to exhibit facet selectivity, and the facet-selective growth only has a limit for effective electron confinement as well. The authors claim that the use of n-octylamine and DPP-Se homogenizes surface energy and suppresses the (111) facet selectivity of ZnSe growth. While this approach is interesting, it should be noted that n-octylamine and DPP-Se have been frequently employed in previous studies, and the reported QY of 92% is not significantly higher than in prior literature. Similarly, the QLED EQE of 23.5% is not unprecedented, with the highest reported value for green InP devices reaching 26.7% (Ref. 10 in this manuscript). However, the reported average peak EQE of around 20% for AM-LED devices is, to the best of my knowledge, unprecedented and represents a noteworthy achievement. The relationship between ZnSe shell thickness and electron confinement has been quantitatively discussed in earlier reports (e.g., ACS Energy Letters, 5 (4), 1316-1327). A more developed interpretation in comparison to those works would strengthen the paper. In addition, evidence beyond lattice parameter analysis is needed to support the claim that facet selectivity governs ZnSe morphology development. In the present study, the QDs have an approximate overall diameter of 7 nm, with an InP core of ~2 nm and a ZnSe thickness of ~2 nm required for >95% electron confinement, plus ~0.5 nm ZnS for further confinement. These dimensions suggest a QD size near the minimal requirement. Considering that in Ref. 10 isotropic ZnSe shells of 2.5–4.5 nm thickness (overall size 8–12 nm) were successfully synthesized and used, and that this was a key improvement in that work, the shell thickness in the present study is actually smaller. Thus, it would be valuable for the authors to further highlight what is truly new here compared to previous results.

The reported PL and EL characteristics of the SEC and WEC QDs—PL: 537 nm (35 nm) and 542 nm (42 nm), EL: 542 nm (43 nm) and 552 nm (52 nm), with Stokes shifts of 5 nm and 10 nm (Supporting Fig. 14)—are indeed consistent with exciton delocalization effects in core/shell QDs, particularly those arising from the relatively light electron. However, this outcome is predictable, and the novelty in this context should be clarified. For example, the SEC and WEC decay times differ by ~1.4×, and the current density difference in EOD devices is ~1.6×. Given that decay times are on the nanosecond scale while current measurements are much slower, it is not immediately clear how these differences can be directly attributed to electron confinement in the QDs. A more detailed explanation would help—for instance, could EL decay time be interpreted in relation to device current density, or could PL changes under applied electric fields provide further insight? I encourage the authors to address such points in more detail.

The most importantly, the paper discusses tuning the InP core surface energy to control ZnSe growth and includes DFT calculations. While this addresses the earliest stage of core–shell interface formation, experimental evidence suggests that the initial ZnSe nucleation on InP surfaces tends to be relatively isotropic, even with the facet selectivity. However, the final morphology (derived from the facet selectivity) emerges more strongly as the ZnSe layer thickens (not on the InP core surface). Thus, the current interpretation may not fully capture the later stages of shell growth. It may be worth considering additional mechanisms and evidence—for example, calculating surface energies for ZnSe facets during shell growth as the shell evolves, or showing possible pathways by which lattice mismatch is accommodated as the shell evolves. These are only suggestions, and better approaches can also be suggested by authors.

The comparison of SEC and WEC QDs—both having relatively low QY due to less effective surface passivation—and the use of their FWHM and decay time differences to argue for electron confinement effects is not entirely convincing. Likewise, the performance difference between the corresponding QLED devices may not be sufficient as sole evidence. Clarifying how the observed effects can be distinguished from exciton or hole confinement (and attributed specifically to electron confinement) would strengthen the argument. Notably, the use of DPP-Se to control Se reactivity has been reported multiple times (e.g., Evans et al., J. Am. Chem. Soc., 132, 10973-10975, 2010), and various alkylamines have also been widely employed (Front. Chem., 6, 2018, doi:10.3389/fchem.2018.00567). A recent paper (Nature Communications, 16, 1945 (2025)) reported that combining DPP-Se with oleylamine allowed size and shape control of ZnTeSe QDs, yielding improved quality under conditions similar to those here (though n-octylamine was not used). These prior results suggest that the ligand combination in the present study is not entirely unprecedented, though the authors have applied it to produce high-quality materials. A clearer mechanistic explanation of how surface energy was tuned in this work would be beneficial.

In conclusion, while the manuscript presents high-quality materials and strong device results, the novelty of the synthesis approach and the proposed underlying mechanism are not yet fully convincing. I believe the work could be publishable in another journal in its current form, but for Nature Communications, additional experimental evidence and mechanistic justification would be necessary to clearly establish the claimed novelty.

Reviewer #3

(Remarks to the Author)

Guo et al. have synthesized strongly electron-confined green InP QDs towards heavy-metal-free ultrahigh-resolution active-matrix displays. They effectively suppress selective growth of shell on the InP (111) facet through introducing n-octylamine and DPP-Se ligands. High-performing QLEDs based on these QDs exhibit a peak EQE of 23.5%, a high luminance and long operational lifetime. Impressively, the high-resolution QLEDs achieve a compelling 8,460 PPI resolution while maintaining average peak EQEs comparable to thin-film QLEDs. I think this article is interesting and meaningful for the next-generation eco-friendly displays based on heavy-metal-free QDs. I recommend publication after the following issues are addressed.

1. In this article, the authors mainly focus on the facet-selective growth of shell layers during the synthesis of InP-based QDs. They proposed that the synergistic effect of octylamine and DPP-Se ligands promotes uniform ZnSe shell growth, thereby effectively passivating surface defects. Although it is supported by morphological and spectroscopic data of InP/ZnSe/ZnS QDs, this article still lacks direct evidence for ligand-mediated ZnSe uniformity without additional ZnS shell. It is essential to

provide comprehensive characterizations of InP/ZnSe QDs with different ligands (e.g. TEM image, PLQY data, etc.) for the validation of their mechanism. Moreover, in Supp. Fig. 7, FTIR and XPS data can only demonstrate the existence of octylamine, thus the authors need to add more data to verify the involvement of DPP ligands. More data or discussions are also needed to further evaluate the effect of octylamine and DPP-Se ligands on controlling facet-selective growth of InP/ZnSe QDs, except XRD and distribution statistics of QD size.

2. It is particularly impressive that the SEC QD-based devices demonstrate sustained efficiency exceeding 20% across an extensive pixel size range (1.5-20 μm). For the high performance in high-resolution devices, the authors attributed it to the controlled assembly achieved through their strategy besides the inherent properties of SEC QDs. They need to provide more detailed discussions to clarify the necessity of their assembly strategy.

3. In Supp. Fig. 15, the authors reported the carrier mobility of their core-shell QDs through hole-only/electron-only devices, which exhibit different device structures of QLED device in Fig. 3a. To ensure reproducibility of this article, they need to provide the detailed methodology in both fabrication and measurement of these devices.

4. Other minor issues should be corrected. For example, the yellow indicators in Fig. 1a are unclear and should be optimized; The color scale in Fig. 2h lacks axis labels; The axis ranges and color bars in Supplementary Figs. 5 and 11 should be unified for better clarity; Unit formats in Supplementary Fig. 11 should be made consistent with the main text; In the figure caption of Supp. Fig. 15, electron-only devices (HODs) is misspelled.

Version 1:

Reviewer comments:

Reviewer #1

(Remarks to the Author)

The authors have thoroughly addressed all the previous concerns and revised the manuscript accordingly. The novelty and significance of the work are now clearly highlighted. I have no further comments, and the manuscript seems suitable for publication in Nature Communications.

Reviewer #2

(Remarks to the Author)

The authors claim that the combined use of n-octylamine and DPP enables reduced steric hindrance of alkyl chains, improved surface conduction, and ultimately the formation of quantum dots with superior properties compared to those synthesized in a conventional tri-octylamine solvent. They further argue that these improvements are experimentally validated and mechanistically supported through surface-energy calculations on both InP cores and ZnSe shell growth. In particular, the authors' efforts to extend their surface-energy analysis to ZnSe shell formation—and to correlate this with uniform shell growth—are technically sound and represent a commendable attempt to support the proposed mechanism. However, despite these strengths, important questions remain regarding the broader impact and originality of the findings. Considering the claimed surface-energy modulation induced by n-octylamine, it is still unclear how the resulting improvements compare with (i) the shell-thickness regimes previously established in the literature, (ii) the commonly reported size distributions of InP/ZnSe(ZnS) QDs, and (iii) the state-of-the-art efficiencies of green InP-based QLEDs. Ligand modifications—whether applied before or after growth—are well-known strategies, and n-octylamine itself is already a reasonable choice within the established ligand library. Thus, it is difficult to identify what fundamentally new concept is being introduced here, especially since the authors' own analysis attributes the strong confinement primarily to uniform shell growth for improved size/shape distribution or any facet-selective growth route.

The authors also argue that the SEC-type QDs yield high performance in unit QD-LED device tests, but such values are comparable to those already reported and do not, on their own, constitute a major conceptual advance suitable for Nature Communications. In contrast, the demonstration of good performance in large scale AM-LEDs—indeed unprecedented—could have been a more compelling point of novelty. If the manuscript had focused more deeply on how the SEC QDs maintain their performance during process scaling and pixelation (connection between QDs and large-area/scale LED performances), this could have significantly strengthened its impact.

Overall, while the manuscript presents scientifically solid work with high-quality materials, strong understanding of the field, and impressive device metrics, the connection between these results and a demonstrably new conceptual advance in AM-LED technology remains insufficiently clear. For these reasons, despite of the appreciation of authors' efforts, I regret to conclude that the work does not meet the novelty and impact threshold required for Nature Communications. I recommend considering submission to a more specialized journal where the strengths of the study will be more appropriately recognized.

Reviewer #3

(Remarks to the Author)

The revised manuscript has satisfactorily addressed all of my concerns. It is now suitable for publication in Nature Communications.

Response to Reviewers' Comments and Revised Details

Manuscript ID: NCOMMS-25-46975

Response to Reviewer 1:

Comment 1: This manuscript presents the synthesis of InP quantum dots (QDs) and their application in QD-light emitting diode (QLED) and active matrix LED display. From the perspective of InP QDs synthesis, the authors utilized different combinations of ligands such as OA:TOP-Se, octylamine:TOP-Se, and octylamine:DPP-Se for the ZnSe shell formation to mitigate its facet-selective growth and promote the homogeneity. The final InP/ZnSe/ZnS QDs exhibited a quasi-spherical shape by using octylamine:DPP-Se combination. They implemented these InP QDs for the fabrication of QLED and an active matrix green display. The QLED exhibited a peak external quantum efficiency (EQE) of 23.50% with a life span of 59,151 h at 100 cd m⁻². However, the findings of this work are not sufficiently novel, making it not suitable for consideration in Nature Communications; therefore, I do not recommend it for publication. The detailed concerns are discussed below:

Response to comment 1: We sincerely appreciate Reviewer 1's time, effort, and thoroughly review on our manuscript. We acknowledge the reviewer's concern on the novelty of this work. However, we would like to emphasize that our novelty focuses on integrating regulation of isotropic shell growth kinetics with asymmetric-wettability-directed assembly to form microscale dense and ordered QD films, which significantly enhance patterning device efficiency, enabling scalable heavy-metal-free displays.

1. Novelty in the regulation of shell growth kinetics.

Although alkylamines and DPP-Se have been reported in the literature, we employed the specific combination of n-octylamine and DPP-Se and explicitly elucidated its role in the active control of homogeneous surface energy and crystallization kinetics. This approach fundamentally differs from conventional methods that rely on high-boiling-point amines such as trioctylamine (TOA) for purely thermodynamic control. Due to its large steric hindrance and weak coordination ability, TOA is primarily used to provide an inert environment rather than achieving precise passivation. In contrast, the smaller steric hindrance and stronger coordination ability of n-octylamine, in synergy with the rapid selenium release from DPP-Se, enable kinetic control over ZnSe shell growth. This approach significantly improves the shell uniformity and enhanced passivation of defect states at the InP/ZnSe interface. This strategy of achieving controllable synthesis of quasi-spherical InP quantum dots (QDs) by modulating surface energy represents a novel method developed based on conventional routes.

2. Towards scalable heavy-metal-free displays: a pathway to overcoming high-resolution QLED efficiency challenges.

A key factor limiting patterning efficiency in high-resolution QLEDs is the formation of non-uniform and porous QD films, which lead to pixel crosstalk, defects, and current leakage (*J. Am. Chem. Soc.* **140**, 8690-8695 (2018)). This issue is particularly severe with morphologically imperfect InP-based QDs. Our work addresses this challenge by combining the highly symmetric QDs with an asymmetric-wettability-mediated assembly strategy. This approach precisely controls solvent evaporation and guides the directional dewetting of the three-phase contact line, thereby suppressing the chaotic flows and enabling the formation of orderly packed, dense, and precisely patterned QD films at the microscale. Consequently, these devices achieve remarkable efficiency (average EQE $\approx 20\%$) even at ultrasmall pixel sizes of 1.5-20 μm . Furthermore, the strategy is compatible with current industrial standard processes, enabling the fabrication of active-matrix QLED display devices. It provides scalable solutions for heavy-metal-free QD display technology, a feature rarely demonstrated in previous works.

In summary, we believe this work achieves significant breakthroughs in multiple aspects, including innovation in the mechanism of the synthetic strategy, realization of quasi-spherical QDs, self-assembly patterning techniques, high-efficiency patterned QLEDs, and active-matrix display demonstrations. These achievements fully meet the criteria of novelty and impact required by *Nature Communications*. We hope these clarifications adequately address Reviewer 1's concerns regarding the novelty of our study.

Comment 2: The authors demonstrated that using octylamine:DPP-Se, they eliminated the facet-selective growth; however, there is a rich literature available where octylamines in combination with TOP-Se have shown the growth of homogenous ZnSe shells on InP QDs, affecting the novelty of this work. For example consider the representative works below: [1] *Nature* **575**, 634–638 (2019); [2] *Nature* **635**, 854–859 (2024).

Response to comment 2: We sincerely appreciate the reviewer for providing insightful comments and highlighting these highly relevant references (*Nature* **575**, 634–638 (2019); *Nature* **635**, 854–859 (2024)). These studies have made indelible contributions to the development of the InP-based QDs and also serve as important references for the homogeneous shell growth of InP-based QDs. However, we would also like to clarify the distinctions of our work.

1. Superior morphological and assembly control - comparison of experimental results highlighting novelty.

Compared with the TEM images of InP-based QDs synthesized with trioctylamine (*Nature* **575**, 634–638 (2019); *Nature* **635**, 854–859 (2024)), the InP/ZnSe/ZnS QDs obtained in this study exhibit a more uniform particle size distribution and higher roundness, thus achieving a higher degree of ordered assembly (**Fig. R1**). Such enhanced morphological control and ordering are critical for achieving efficient self-assembly and improving the performance of display devices (*Nat Commun* **16**, 4257 (2025); *Nat Commun* **16**, 7643 (2025)).

2. Mechanistic distinctions supported by theoretical calculations.

Through binding energy calculations (**Fig. R1**), we confirmed that n-octylamine exhibits a significantly higher binding energy to the QD surface compared to oleylamine and TOA, which is closely associated with its smaller steric hindrance and stronger coordination capability. The rapid selenium release kinetics of DPP-Se, in synergy with n-octylamine, substantially altering the energy landscape of the core surface during the early stages of shell growth, thereby effectively eliminating facet-selective growth.

3. Distinct core mechanism from conventional thermodynamic control in the literature.

In previous studies, the combination of alkylamines (particularly TOA) and TOP-Se primarily facilitated shell growth through thermodynamic control in a high-boiling-point environment. Owing to its substantial steric hindrance and weak coordination ability, TOA exhibits limited effectiveness in passivating small facets. In contrast, our study employs a specific combination of n-octylamine and DPP-Se, and explicitly demonstrates its role in kinetic regulation and targeted facet passivation. This approach represents not merely a solvent or ligand substitution, but a novel mechanism enabling isotropic growth through precise surface energy modulation.

Therefore, our work presents a rational ligand engineering strategy that distinctly differs from conventional thermodynamic approaches, enabling precise kinetic control over shell morphology.

[FIGURE REDACTED]

Fig. R1. **a**, TEM images of green InP/ZnSe/ZnS QDs (*Nature* **635**, 854–859 (2024)). **b**, STEM images of red InP/ZnSe/ZnS QDs (*Nature* **575**, 634–638 (2019)). **c**, TEM images of InP/ZnSe/ZnS QDs in this work.

Fig. R2. Adsorption model and the binding energy (ΔE) of **a**, tri-octylamine, **b**, oleylamine and **c**, n-octylamine on (111) facet of InP.

Comment 3: In the XRD analysis (Fig. 1h), the authors claimed that the reduced (111) peak intensity in SEC QDs shows suppressed ZnSe growth on the (111) facet. However, the XRD peak intensities are also influenced by multiple factors, including crystal orientation, size effects, etc.

Response to comment 3: We sincerely appreciate the reviewer's constructive feedback. The use of XRD as an auxiliary method for identifying facet-selective growth has been widely validated in numerous studies (e.g., *Angew. Chem. Int. Ed.* **59**, 5385 (2020); *Nat Commun.* **15**, 5484 (2024)). Moreover, we did not rely solely on the reduction in the (111) peak intensity as independent evidence; instead, we combine it with EDS mapping and TEM images, as shown in **Supplementary Fig. 5** and **Figs. R3 and 4**. While WEC QDs exhibit (111) facet-preferential growth leading to a tetrapod-like morphology, the SEC QDs display a uniform spherical shape with multiple exposed facets. The cross-validation of these multiple results supports our conclusion that the reduced (111) peak intensity in SEC QDs originates from suppressed growth of ZnSe on the (111) facet.

Fig. R3. Morphology of QDs synthesized with the combination of OA and TOP-Se. **a**, A high-angle annular dark field (HAADF) image. **b**, Energy dispersive spectroscopy (EDS) elemental mapping of Se elements from several WEC InP/ZnSe/ZnS QDs. **c**, Energy dispersive spectroscopy (EDS) elemental mapping of Se (green), Zn (blue) and S (red) elements from several WEC InP/ZnSe/ZnS QDs. All the scale bars are equal to 5 nm.

Fig. R4. HRTEM images and inverse Fourier transform images of QDs synthesized with the combination of (a) OA and TOP-Se and (b) n-octylamine and DPP-Se. The scale bars in the HRTEM images and the FFT images are 5 nm and 1 nm, respectively.

To enhance the quality of our manuscript, we have added relevant discussions in the revised version, as detailed below:

Page 6, Lines 16-27: Energy-dispersive spectroscopy element mapping demonstrates the distribution of Se in QDs synthesized with OA and TOP-Se closely aligns with the pod-like morphology, demonstrating more pronounced facet-selective growth as the ZnSe shell thickens (Supplementary Fig. 6)^{32,33}. In contrast, the InP/ZnSe/ZnS QDs synthesized using n-octylamine with DPP-Se possess a highly symmetric core/multishell structure with uniform ZnSe shell growth (Supplementary Fig. 7). In addition, X-ray diffraction in Fig. 1h also shows that the (111) facet diffraction peak

intensity of InP/ZnSe/ZnS QDs synthesized with a combination of n-octylamine and DPP-Se ligands is the lowest, which is consistent with the high-resolution TEM images analysis (Supplementary Fig. 8). These observations collectively indicate that our ligand combination successfully suppresses the selective growth of ZnSe on the (111) facet³².

Comment 4: While Figures 2f-h demonstrate superior photostability of SEC QDs through PLQY decay under UV irradiation, solution-phase PL intensity, and temperature-dependent PL spectra, the authors provided no mechanistic explanation linking these observations to enhanced electron confinement.

Response to comment 4: We sincerely thank the reviewer for this insightful and constructive comment. We agree that clarifying the mechanistic link is essential. The enhanced electronic confinement achieves the superior photostability of SEC QDs through the following pathways:

1. The enhanced electron confinement alleviating the surface defect recombination.

The uniform shell growth of SEC QDs significantly suppressed the delocalization of electron wavefunctions toward the shell surface. This effectively reduced PL quenching and exciton-surface coupling caused by electrons being captured by surface traps (*Adv. Funct. Mater.* **30**, 2005303 (2020); *Angew. Chem. Int. Ed.* **64**, e202420421 (2025)). The higher potential barrier for electrons to escape makes it more difficult for the energy provided by ultraviolet light or temperature to drive electrons to the surface, significantly enhancing the stability.

2. Enhanced electron confinement increases the exciton binding energy and boosts radiative recombination.

The effective confinement of electrons maximizes the spatial overlap between electron and hole wavefunctions. Consequently, the Coulombic interaction between the electron-hole pair is significantly enhanced, leading to an increase in the exciton binding energy, making the exciton more stable against dissociation (*Angew. Chem. Int. Ed.* **63**, e202318777 (2024)). As a result, a higher fraction of photogenerated excitons in SEC QDs preferentially recombine radiatively, thereby directly manifesting as the stronger PL intensity and superior stability observed under various harsh conditions.

To enhance the quality of our manuscript, we have revised relevant discussions in the manuscript, as detailed below:

Page 8, Lines 2-13: Additionally, the SEC QDs achieved a PLQY of 92.28%, significantly surpassing the 64.46% of the WEC QDs, and exhibited superior photostability under UV irradiation (Fig. 2f and Supplementary Table 3). This enhanced

performance is attributed to the superior electron confinement, which effectively isolates electrons from surface traps and thereby significantly reducing non-radiative recombination via exciton-surface coupling^{40,41}. Concurrently, this confinement promotes greater overlap of electron and hole wavefunctions, which enhances the exciton binding energy and boosts the radiative recombination rate. This dual mechanism—promoting radiative recombination while suppressing non-radiative decay—is further corroborated by the higher PL intensity of the SEC QDs at an equivalent optical density (Fig. 2g) and their exceptional stability in temperature-dependent PL measurements (Fig. 2h and Supplementary Fig. 12).

Comment 5: Regarding the device performance, the authors attributed 107.5-fold operational lifetime enhancement in SEC QD device exclusively to suppressed electron leakage into the HTL via strong electron confinement lacks mechanistic validation. There could be other parameters involved, such as different ligands that could passivate the QDs' surface and affect their overall stability under electric field.

Response to comment 5: We thank the reviewer for this constructive feedback. We agree that device operational stability is influenced by multiple factors. However, in our work, we propose that the primary factor contributing to the significant enhancement of device lifetime is the suppressed electron leakage achieved via stronger electron confinement. It must be admitted that we did introduce new ligands when synthesizing SEC QDs. To address the reviewer's concern, we synthesized InP/ZnSe/ZnS QDs with a thinner ZnSe shell (~1 nm) compared to the SEC QDs (~2 nm) also using n-octylamine and DPP-Se. We further fabricated devices based on these QDs and characterized their operational lifetime. The results show that the lifetime of this QD-based device is only $T_{50} \approx 3457$ h at 100 cd m^{-2} , exhibiting merely a 6.3-fold enhancement relative to the WEC QDs. Although these QDs share the same ligands as the SEC QDs, the thinner ZnSe shell fails to achieve strong electron confinement (*ACS Energy Letters*, **5** (4), 1316-1327), which still leads to electron leakage and poor device lifetime. This result clearly indicates that while the ligands themselves contribute to some extent, the key factor responsible for the 107.5-fold enhancement in device lifetime is the strong electron confinement.

Fig. R5. Lifetime measurements of QLED devices based on InP/ZnSe/ZnS QDs synthesized using n-octylamine and DPP-Se with a thinner ZnSe shell (~1 nm). **a**, Luminance and time dependency characteristics curves of devices based on InP/ZnSe/ZnS QDs synthesized using n-octylamine and DPP-Se with a thinner ZnSe shell (~1 nm). **b**, Extrapolation of accelerating factor (n) for the lifetime estimation.

To enhance the quality of our manuscript, we have added relevant discussions in the revised version, as detailed below:

Page 11, Lines 12-19: To further rule out potential interference from different ligands, we synthesized InP/ZnSe/ZnS QDs featuring a thinner ZnSe shell (~1 nm) compared to that of the SEC QDs (~2 nm), while maintaining the same ligands of n-octylamine and DPP-Se. The corresponding devices exhibited a lifetime of only $T_{50} \approx 3457$ h at 100 cd m^{-2} (Supplementary Fig. 23), which mainly attributed to the insufficient electron confinement of the thinner ZnSe shell. The result further confirmed that strong electron confinement is the key factor underlying the significant improvement in device lifetime.

Comment 6: The authors provided no experimental evidence demonstrating how electron leakage initiates HTL degradation, nor did they consider factors like interfacial defect generation across different layers.

Response to comment 6: We sincerely thank the reviewer for this valuable comment. To address this comment, we fabricated QLEDs with identical structures using two types of QDs and operated them at a constant current density of 2.0 mA cm^{-2} for 20 h. Subsequently, the top layers were carefully removed in a nitrogen-filled glovebox to expose the PF8Cz layer (*Nat. Commun.* **14**, 7785 (2023)). The results revealed that in devices based on WEC QDs, the PF8Cz absorption exhibited a red shift with a pronounced decrease in intensity, accompanied by enhanced low-energy emission, whereas in devices based on SEC QDs, no significant changes in PF8Cz were observed

(**Fig. R6b-d**). To confirm the origin of the significant degradation, we conducted XPS measurements on the aged HTL, the results showed that the O 1s spectrum of HTL in the device based on WEC QDs presented an additional peak assigned to C=O at 288.1eV, while this did not occur in the corresponding device based on SEC QDs (**Fig. R6e, f**). This observation demonstrated the electro-oxidation of fluorene units in PF8Cz to fluorenone group induced by leakage electrons. The significantly enhanced green band emission in the emission spectrum of PF8Cz after electrical aging of the device based on WEC QDs also proves this point (**Fig. R6g**). (*Adv. Mater.* **36**, 2309123 (2024); *J. Phys. Chem. Lett.* **11**, 4649 (2020)). The generation of fluorenone defects hindered hole transport, and their strong electron-withdrawing effect reduces the band gap, corresponding to a redshift in the absorption spectrum.

Fig. R6. The degradation mechanism of the HTL. **a**, Schematics of exposing the surfaces of PF8Cz in the QLEDs by removing all the top layers. **b, c**, Absorption spectra of PF8Cz HTLs in QLEDs based on **(b)** WEC QDs and **(c)** SEC QDs before and after operation at 2.0 mA cm^{-2} for 20 h. **d, e**, XPS spectra of PF8Cz HTLs in QLEDs based on **(d)** WEC QDs and **(e)** SEC QDs after electrical aging at 2.0 mA cm^{-2} for 20 h. **f**, PL spectra of PF8Cz in QLEDs based on WEC QDs before and after operation at 2.0 mA cm^{-2} for 20 h. **g**, The mechanism of electron leakage initiating HTL degradation by the generating fluorenone defect sites in PF8Cz.

To enhance the quality of our manuscript, we have added relevant discussions in the revised version, as detailed below:

Page 10, Lines 25-28, Page 11, Lines 1-12: We attributed the substantially improved operational lifetime to strong electron confinement of SEC QDs, which suppressed electron leakage to HTL and mitigates HTL degradation. To confirm the impact of charge leakage on HTL degradation, we fabricated QLEDs with identical structures using two types of quantum dots and operated them at a constant current density of 2.0 mA cm⁻² for 20 h. The top layers were then removed to measure the absorption and photoluminescence spectra of PF8Cz before and after device operation (Supplementary Fig. 22a). The results reveal that after the same operational time, the absorption of PF8Cz in devices based on WEC QDs shows marked redshift and decrease in intensity, accompanied by a significantly enhancements in the lower-energy emissions (Supplementary Fig. 22b-d). The emergence of additional peak assigned to C=O at 288.1 eV in C 1s demonstrated that this degradation can be attributed to the electro-oxidation of fluorene units in PF8Cz to fluorenone group induced by leakage electrons (Supplementary Fig. 22e-g)^{43,47}. In contrast, the HTL in devices based on SEC QDs shows no evident degradation, suggesting that enhanced electron confinement helps to mitigate HTL degradation, thereby improving the operational stability of the devices.

Page 15, Lines 20-25: The peel-off experiments were carried out on unencapsulated QLEDs in a nitrogen-filled glovebox to expose the PF8Cz layer. The Al cathode was first detached using polyimide tape. Then, the ZnMgO layer was washed away by spin-coating 30 μ L of an acetic acid-ethanol solution (0.8% by volume) at 2000 rpm for 30 seconds, followed by the same process using 30 μ L of pure ethanol. The QD layer was removed by spin-coating 30 μ L n-octane at 2000 rpm for 30 seconds.

Comment 7: The attribution of the 430 nm parasitic peak in WEC device solely to electron leakage-induced recombination in the HTL requires further investigations. To strengthen this claim the EL spectra from PF8Cz-based control device will be helpful to verify whether its intrinsic emission aligns with the 430 nm feature.

Response to comment 7: We are sincerely grateful to the reviewer for this constructive feedback. Following the reviewer's suggestion, we fabricated a PF8Cz-based control device with the structure ITO/PEDOT:PSS/PF8Cz/ZnMgO/Al and systematically characterized its EL spectrum. Their emission spectra under different driving voltages are shown in **Fig. R7a**. The intrinsic emission of PF8Cz exhibits two peaks at 407 nm and 435 nm, align consistently with those from devices based on WEC QDs (**Fig. R7b**). This agreement indicates that the parasitic emission indeed originates from electrons leaking through the QD layer into the PF8Cz layer, where radiative recombination occurs.

Fig. R7. HTL emission caused by highly delocalized electron leakage from the InP QD layer to the HTL, where they recombine with holes. a, EL spectrum of PF8Cz-based control device (with a structure of ITO/PEDOT:PSS/PF8Cz/ZnMgO/Al). **b,** EL spectra of QLED based on WEC InP/ZnSe/ZnS QD films under different voltages. (Inset: magnified spectra marked by the rectangle showing parasitic emission).

To strengthen our manuscript, we have revised our manuscript as follows:

Page 9, Lines 15-20: Due to the strong electron delocalization driven by the electric field, the EL spectrum of the QLED based on WEC QD films showed a parasitic emission peak (see Fig. 3c and Supplementary Fig. 18a), which is consistent with the intrinsic EL spectrum of PF8Cz (Supplementary Fig. 18b). This is attributed to highly delocalized electron leakage from the InP QD layer to the HTL, where they recombine with holes in the HTL^{21,25}.

Comment 8: The maximum EQE of the QLED reported is 23.50% which is lower than the recent reported EQE (26.68%) for the green InP QLED, why? (*Nature* **635**, 854–859 (2024))

Comment 10: The T_{50} at 100 cd m^{-2} for QLED based on SEC InP QDs reached 59,151 h only, whereas the recent best reported case for green InP QLED demonstrated a T_{95} of 89,900 h under similar brightness conditions. (*Nature* **635**, 854–859 (2024))

Response to comment 8 and 10: We sincerely thank the reviewer for their insightful comments. The remarkable work in *Nature* **635**, 854–859 (2024) indeed offers a new paradigm and direction for research on InP-based QLEDs, marking a significant milestone in the development of their performance. We would like to clarify, however, that our work specifically addresses the issue of facet-selective growth of ZnSe shells by proposing a ligand engineering strategy that significantly enhances electron

confinement. This approach is consistent with the perspective presented in *Nature* **635**, 854–859 (2024) that low efficiency of green InP-based QD-LEDs originates from the low electron concentration. Although the peak EQE (23.5%) and operational stability ($T_{50}@100 \text{ cd m}^{-2} = 59,151 \text{ h}$) reported in our study do not surpass the record-setting values, they still rank among the highest values reported to date. Furthermore, our work demonstrates the integration of high device performance with high-resolution patterning capability. The successful application in active-matrix displays provides a scalable template for high-performance, heavy-metal-free patterned QLED devices.

In the revised manuscript, we have explicitly highlighted this contribution in the Introduction (Page 3, Lines 14–17) and compared our results with the state-of-the-art performance records in the literature to underscore the unique value and application potential of our work:

Page 3, Lines 14-16: Recent studies have clearly demonstrated that the inferior performance of green InP-based QLEDs is primarily due to insufficient electron concentration, a key finding that deepens the understanding of green InP-based QLEDs¹⁰.

Comment 9: The FWHM of the EL spectrum increased to 43 nm, much broader than its PL spectrum, which is 35 nm. No attention is paid to this factor, like what caused such broadening, as it affects the color purity.

Response to comment 9: We sincerely thank the reviewer for raising this critical point. We have compiled and analyzed the full width at half maximum (FWHM) of both PL and EL spectra from published studies on green InP-based QLEDs (**Table R1**). The data indicate that significant spectral broadening (often exceeding 10 nm) commonly occurs from PL to EL. The observed broadening of the EL spectrum can be attributed to the electric field-induced Stark effect: under an applied electric field, insufficient confinement leads to a spatial shift of the wave function toward the shell surface and enhanced coupling with trap states (*Nat Rev Electr Eng* **1**, 412–425 (2024); *Light Sci. Appl.* **11**, 162 (2022)). This issue is particularly pronounced in InP-based QDs due to the smaller electron effective mass and weaker electronic confinement by the shell compared to Cd-based QDs. In this context, although an 8 nm broadening is observed in our devices, it represents an improvement compared to both control devices and the majority of values reported in the literature.

Table R1. PL and EL FWHM of green InP-based QDs reported in the literature.

QD structure	PL FWHM (nm)	EL FWHM (nm)	Broadening of FWHM (nm)	EQE (%)	Ref.
InP/ZnSe _{0.7} S _{0.3} /ZnS	35	45	10	15.2	1
InP/ZnSe/ZnS	36	46	10	13.8	2
InP/ZnSe/ZnS/BDA	35	45	10	16.3	3
InP/ZnSe/ZnS	40	50	10	6.2	4
InP/ZnSe/ZnS	35	38-39	4	26.68	5
InP/ZnS	43	48	5	6.75	6
InP/Q-ZnSe/ZnS	36	44	8	10.6	7
InP/ZnSeS/ZnS	36	42	6	5.56	8
InP/ZnSeS/ZnS	45	55	10	7.06	9
InP/ZnSe/ZnS	43	>50	17	15	10
InP/ZnSe/ZnSe _x S _{1-x} /ZnS	39	~50	11	15.4	11
InP/ZnSe/ZnS	35	43	8	23.5	Our work

References

1. Yu, P. et al. Highly efficient green InP-based quantum dot light-emitting diodes regulated by inner alloyed shell component. *Light Sci. Appl.* **11**, 162 (2022).
2. Wu, Q. et al. Bridging Chloride Anions Enables Efficient and Stable InP Green Quantum-Dot Light-Emitting Diodes. *Adv. Optical Mater.* **11**, 2300659(2023).
3. Chao, WC. et al. High efficiency green InP quantum dot light-emitting diodes by balancing electron and hole mobility. *Commun. Mater.* **2**, 96 (2021).
4. Sun, Z. et al. Suppressing the Cation Exchange at the Core/Shell Interface of InP Quantum Dots by a Selenium Shielding Layer Enables Efficient Green Light-Emitting Diodes. *ACS Appl. Mater. Interfaces* **14**, 15401–15406 (2022).
5. Bian, Y. et al. Efficient green InP-based QD-LED by controlling electron injection and leakage. *Nature* **635**, 854-859 (2024).

6. Wang, L. et al. Modified Charge Injection in Green InP Quantum Dot Light-Emitting Diodes Utilizing a Plasma-Enhanced NiO Buffer Layer. *J. Phys. Chem. C* **128**, 3985–3993 (2024).
7. Wu, Q. et al. Quasi-Shell-Growth Strategy Achieves Stable and Efficient Green InP Quantum Dot Light-Emitting Diodes. *Adv. Sci.* **9**, 2200959 (2022).
8. Zhao, H. One-Pot Synthesis of InP Multishell Quantum Dots for Narrow-Bandwidth Light-Emitting Devices. *ACS Appl. Nano Mater.* **6**, 3797-3802 (2023).
9. Liu, P. et al. Green InP/ZnSeS/ZnS Core Multi-Shelled Quantum Dots Synthesized with Aminophosphine for Effective Display Applications. *Adv. Funct. Mater.* **31**, 2008453 (2021).
10. Gao, P. et al. Efficient InP Green Quantum-Dot Light-Emitting Diodes Based on Organic Electron Transport Layer. *Adv. Opt. Mater.* **10**, 2202066 (2022).
11. Luo, Y. et al. Suppression of Interfacial Oxidation in Core/Shell InP Quantum Dots through Solvent Assisted Core-Etching Strategy for Efficient Green Light-Emitting Diodes. *Nano Lett.* **25**, 593-599 (2025).

Response to Reviewer 2:

Comment 1: The authors report that the combined use of n-octylamine and DPP-Se enables isotropic ZnSe shell growth on InP cores, thereby enhancing electron confinement. They achieve a QY above 92%, an FWHM as narrow as 35 nm, a QLED EQE exceeding 23.5%, and a maximum brightness above 140,000 cd m⁻². They also demonstrate an asymmetric wettability-mediated assembly strategy for QD arrays with a resolution of 8,460 ppi, integrated into an AM-display prototype.

It is known that ZnSe tends to exhibit facet selectivity, and the facet-selective growth only has a limit for effective electron confinement as well. The authors claim that the use of n-octylamine and DPP-Se homogenizes surface energy and suppresses the (111) facet selectivity of ZnSe growth. While this approach is interesting, it should be noted that n-octylamine and DPP-Se have been frequently employed in previous studies, and the reported QY of 92% is not significantly higher than in prior literature. Similarly, the QLED EQE of 23.5% is not unprecedented, with the highest reported value for green InP devices reaching 26.7% (Ref. 10 in this manuscript). However, the reported average peak EQE of around 20% for AM-LED devices is, to the best of my knowledge, unprecedented and represents a noteworthy achievement.

Response to comment 1: We sincerely thank the reviewer 2 for the time and effort invested in providing constructive feedback and for acknowledging the unprecedented achievement in our AM-LED devices. We would also like to clarify the distinctions and contributions of our work.

1. Distinct ligand combinations and mechanisms of shell growth regulation.

In previous studies on InP-based QDs (*Nature* **575**, 634-638 (2019); *Nature* **635**, 854-859(2024)), particularly trioctylamine (TOA) was commonly used among alkylamines, primarily leverages its high boiling point for thermodynamic control to facilitate shell growth. Due to its large steric hindrance and weak coordination ability, TOA is primarily used to provide an inert environment rather than achieving precise passivation. In contrast, our work precisely utilizes the stronger binding and smaller steric profile of n-octylamine, synergized with the rapid Se release kinetics of DPP-Se, to achieve kinetic control over shell growth. This strategy of achieving controllable synthesis of quasi-spherical InP-based QDs by modulating surface energy represents a novel method developed based on conventional routes.

2. Overall innovation and application advancement.

While the PLQY (92%) and peak EQE (23.5%) reported in our study do not surpass the record values, they still rank among the highest values reported to date. Numerous

established studies have demonstrated that a high PLQY, while beneficial, is not the sole determinant of high electroluminescence performance (*Nat. Rev. Electr. Eng.* **1**, 412–425 (2024)). Efficient charge injection and balanced transport are equally critical (*Chem. Commun.* **55**, 3501–3504 (2019); *Nano Lett.* **21**, 7252–7260 (2021)), which our devices successfully achieve. Furthermore, not only have we made progress in synthesizing high-performance InP QDs, but we have also employed an asymmetric wettability-mediated self-assembly strategy to achieve orderly and precise patterning of QDs at the micrometer scale, while preserving their excellent electroluminescent performance (EQE > 20%). The strategy is compatible with current industrial standard processes, enabling the fabrication of active-matrix QLED display devices. It provides scalable solutions for heavy-metal-free QD display technology, a feature rarely demonstrated in previous works.

Comment 2: The relationship between ZnSe shell thickness and electron confinement has been quantitatively discussed in earlier reports (e.g., *ACS Energy Letters*, **5** (4), 1316-1327). A more developed interpretation in comparison to those works would strengthen the paper.

Response to comment 2: We sincerely thank the reviewer 2 for providing constructive feedback. Prior literatures have provided useful guidance regarding the relationship between ZnSe shell thickness and electron confinement. For green-emitting InP cores, it has been suggested that a shell thickness of at least 1.47 nm is required to remain 95% probability of electron below the ZnSe shell. However, such models are often simply based on isotropic spherical morphology and do not take into account the irregular morphologies caused by the selective growth of ZnSe. In this case, inevitably, a much thicker ZnSe shell is often required.

In contrast, our work achieves oriented ZnSe shell growth through surface energy modulation, which enhances electron confinement even at a shell thickness of ~2 nm, well within the range predicted by the referenced study (*ACS Energy Letters*, **5** (4), 1316-1327). Unlike conventional strategies that rely primarily on increasing shell thickness, we introduce surface energy modulation as a new dimension for achieving strong electron confinement in relatively thin shells. The resulting QDs, which combine isotropic morphology with strong electron confinement, are better suited for our assembly strategy. This enables the formation of more uniform and defect-free patterned films, ultimately leading to improved performance in high-resolution displays.

We thank you for recommending this literature, which have been included in our revised manuscript. To strengthen our manuscript, we have included related discussion

as:

Page 3, Lines 24-25: However, ZnSe tends to show facet-selective growth on InP cores, which leads to stronger electron delocalization that deviates from the spherical ZnSe shell model¹⁶.

Page 26, Lines 9-11: 16. Jang, E. et.al. Environmentally Friendly InP-Based Quantum Dots for Efficient Wide Color Gamut Displays. *ACS Energy Lett.* **5**, 1316-1327 (2020).

Comment 3: In addition, evidence beyond lattice parameter analysis is needed to support the claim that facet selectivity governs ZnSe morphology development.

Response to comment 3: We sincerely thank you for this insightful comment. In the revised manuscript, we have added energy-dispersive X-ray spectroscopy (EDS) elemental mapping of the tetrapod-shaped WEC QDs (**Fig. R8**). The distribution of Se closely aligns with the pod-like morphology, demonstrating pronounced facet-selective growth. This result further supports the argument that morphological evolution of the ZnSe shell is dominated by facet-selective growth.

Fig. R8. Morphology of QDs synthesized with the combination of OA and TOP-Se. **a**, A high-angle annular dark field image. **b**, EDS mapping of Se elements. **c**, EDS mapping of Se (green), Zn (blue) and S (red) elements. All the scale bars are equal to 5 nm.

To enhance the quality of our manuscript, we have added relevant discussions in the revised version, as detailed below:

Page 6, Lines 16-22: Energy-dispersive spectroscopy element mapping demonstrates the distribution of Se in QDs synthesized with OA and TOP-Se closely aligns with the pod-like morphology, demonstrating more pronounced facet-selective growth as the ZnSe shell thickens (Supplementary Fig. 6)^{32,33}. In contrast, the InP/ZnSe/ZnS QDs synthesized using n-octylamine with DPP-Se possess a highly symmetric core/multishell structure with uniform ZnSe shell growth (Supplementary Fig. 7).

Comment 4: In the present study, the QDs have an approximate overall diameter of 7 nm, with an InP core of ~2 nm and a ZnSe thickness of ~2 nm required for >95% electron confinement, plus ~0.5 nm ZnS for further confinement. These dimensions suggest a QD size near the minimal requirement. Considering that in Ref. 10 isotropic ZnSe shells of 2.5 - 4.5 nm thickness (overall size 8 - 12 nm) were successfully synthesized and used, and that this was a key improvement in that work, the shell thickness in the present study is actually smaller. Thus, it would be valuable for the authors to further highlight what is truly new here compared to previous results.

Response to comment 4: We sincerely thank the reviewer for these insightful comments. We respectfully emphasize that, the true novelty of our work lies not in maximizing shell thickness, but in achieving precise control over the shell morphology and interface quality at this critical, size-constrained regime. While Ref. 10 achieved impressive results with larger QDs, they are not strictly isotropic and did not form long-range ordered arrangements or more uniform emissions. Our work demonstrates that a relatively thinner (~2 nm), ultra-uniform shell is sufficient to yield superior and consistent optoelectronic performance enabled by strong electron confinement through precise surface energy regulation from the earliest stages. In contrast to the thick-shell strategies emphasized in previous studies, we have expanded the approach to achieving strong electron confinement by introducing surface energy modulation in thin shells as a novel dimension. In this case, a thicker ZnSe shell layer was no longer needed. A larger size would instead lead to the accumulation of internal defects, resulting in a decrease in fluorescence intensity (*Adv. Energy Mater.* **15**, 2400148 (2025); *J. Phys. Chem. Lett.* **6**, 1559–1562 (2015); *Nano Letters* **3**, 799–803 (2003)).

Comment 5: The reported PL and EL characteristics of the SEC and WEC QDs—PL: 537 nm (35 nm) and 542 nm (42 nm), EL: 542 nm (43 nm) and 552 nm (52 nm), with Stokes shifts of 5 nm and 10 nm (Supporting Fig. 14)—are indeed consistent with exciton delocalization effects in core/shell QDs, particularly those arising from the relatively light electron. However, this outcome is predictable, and the novelty in this context should be clarified. For example, the SEC and WEC decay times differ by $\sim 1.4\times$, and the current density difference in EOD devices is $\sim 1.6\times$. Given that decay times are on the nanosecond scale while current measurements are much slower, it is not immediately clear how these differences can be directly attributed to electron confinement in the QDs. A more detailed explanation would help—for instance, could EL decay time be interpreted in relation to device current density, or could PL changes under applied electric fields provide further insight? I encourage the authors to address

such points in more detail.

Response to comment 5: We thank the reviewer for raising these insightful points. The correlation between nanosecond-scale PL lifetime and macroscopic current density arises from their shared dependence on electron confinement. The extended PL lifetime observed in SEC QDs (84.83 ns vs. 57.95 ns, a difference of approximately 1.4 \times) stems from reduced electron wavefunction leakage, which suppresses non-radiative recombination pathways by minimizing interactions with defect states at the QD surface. In contrast, WEC QDs exhibit weaker electron confinement, leading to enhanced non-radiative recombination through surface trap states. Although WEC QD-based devices show a significantly higher current density (1.6 \times) in EOD compared to SEC-based devices, their device efficiency is markedly lower. This reduction is similarly attributed to leakage currents resulting from insufficient electron confinement, which affects the charge balance.

Furthermore, we have supplemented electric-field-dependent PL measurements in the revised manuscript. WEC QDs exhibit pronounced PL quenching (60 %) with a red shift of \sim 5 nm, showing a high sensitivity to the electric field owing to the field-induced electron dissociation (*Adv. Mater.* **36**, 2309123 (2024)). In contrast, SEC QDs were less sensitive to electric fields and demonstrate minimal PL quenching (14 %) with a red shift of \sim 2 nm, indicating strong confinement.

Fig. R9. PL spectra under different applied electric fields of (a) WEC QDs and (b) SEC QDs in EOD (ITO/ZnMgO/QD/ZnMgO/Al).

To enhance the quality of our manuscript, we have added relevant discussions in the revised version, as detailed below:

Page 9, Lines 7-15: The EL spectra of the SEC QDs exhibit a redshift of approximately 5 nm compared to the PL spectra, whereas the EL spectra of the WEC QDs shows a 10 nm redshift along with significant broadening (Supplementary Fig. 16). Furthermore, in electric-field-dependent PL measurements, the WEC QDs showed a higher

sensitivity to the electric field and demonstrate more pronounced PL quenching and red shift with increasing voltage compared to the SEC QDs (Supplementary Fig. 17). These observations indicate a weaker Stark effect in the SEC QDs under an electric field, which can be attributed to strong electron confinement that suppresses the delocalization of electron wavefunctions toward the shell surface^{2,43}.

Comment 6: The most importantly, the paper discusses tuning the InP core surface energy to control ZnSe growth and includes DFT calculations. While this addresses the earliest stage of core-shell interface formation, experimental evidence suggests that the initial ZnSe nucleation on InP surfaces tends to be relatively isotropic, even with the facet selectivity. However, the final morphology (derived from the facet selectivity) emerges more strongly as the ZnSe layer thickens (not on the InP core surface). Thus, the current interpretation may not fully capture the later stages of shell growth. It may be worth considering additional mechanisms and evidence—for example, calculating surface energies for ZnSe facets during shell growth as the shell evolves, or showing possible pathways by which lattice mismatch is accommodated as the shell evolves. These are only suggestions, and better approaches can also be suggested by authors.

Response to comment 6: We sincerely thank the reviewer for this insightful and constructive comment. In response to Reviewer 2's valuable suggestions, we supplemented the morphological evolution analysis of the ZnSe shell in the later stage of growth. The distribution of Se elements in the EDS element mapping of WEC QDs shows that without any surface modifiers, the anisotropic growth of ZnSe will indeed become more pronounced as the ZnSe shell thickens (**Fig. R10**). To better understand the role of surface energy in the later stages of shell growth, we have now calculated the surface energies of the (111), (100) and (110) facets of ZnSe as the shell evolves (**Fig. R11** and **Table R2**). The results show that the (111) facet exhibits a significantly higher surface energy compared to the (100) and (110) facets (*Angew. Chem. Int. Ed.* **59**, 5385 (2020); *Nat. Commun.* **16**, 2450 (2025)). But when n-octylamine and DPP-Se are adopted, they preferentially adsorb onto the ZnSe (111) facet and markedly reduce its surface energy. These results suggest that the ligands we adopted do not merely influence the initial shell growth on the InP core surface but continue to play a crucial role throughout the entire ZnSe shell growth process. By dynamically stabilizing high-energy facets, the ligands help balance growth rates across the three facets, resulting in a more uniform QD morphology as the shell thickens.

Fig. R10. Morphology of QDs synthesized with the combination of OA and TOP-Se. **a**, A high-angle annular dark field (HAADF) image. **b**, Energy dispersive spectroscopy (EDS) elemental mapping of Se elements from several WEC InP/ZnSe/ZnS QDs. **c**, Energy dispersive spectroscopy (EDS) elemental mapping of Se (green), Zn (blue) and S (red) elements from several WEC InP/ZnSe/ZnS QDs. All the scale bars are equal to 5 nm.

Fig. R11. Slab model of QD facets with ZnSe epitaxial layer. Slab model for the (111), (100), and (110) facets of InP substrate with a monolayer ZnSe epitaxial layer without ligands (top) and with ligands (bottom).

Facets	surface energy ($\text{eV } \text{\AA}^{-2}$)	Post-adsorption surface energy ($\text{eV } \text{\AA}^{-2}$)	Formation energy (eV)
(111)	0.509	0.288	-51.19
(100)	0.303	0.172	-17.57

(110)	0.103	0.100	-0.52
-------	-------	-------	-------

Table R2. DFT formation energies of octylamine and DPP-Se on (100), (110) and (111) facets of InP with ZnSe epitaxial layer.

Comment 7: The comparison of SEC and WEC QDs—both having relatively low QY due to less effective surface passivation—and the use of their FWHM and decay time differences to argue for electron confinement effects is not entirely convincing. Likewise, the performance difference between the corresponding QLED devices may not be sufficient as sole evidence. Clarifying how the observed effects can be distinguished from exciton or hole confinement (and attributed specifically to electron confinement) would strengthen the argument.

Response to comment 7: We sincerely thank the reviewer for this insightful and critical comment. For InP/ZnSe QDs, the valence band offset is larger than the conduction band offset (*Nat. Nanotechnol.* **18**, 993–999 (2023)), resulting in stronger hole confinement and relatively weaker electron confinement. Additionally, electrons have a lighter effective mass compared to holes, which have a heavier effective mass (*Nat. Rev. Electr. Eng.* **1**, 412–425 (2024); *Nature* **635**, 854–859 (2024)). As a result, epitaxial shell growth significantly influences the confinement strength and wave function leakage of electrons, while having minimal impact on the confinement and wave function behavior of holes. For instance, in the case of SEC QDs coated with a ZnSe shell, the VD value of the first excitonic absorption peak increases compared to that of the core, indicating improved size uniformity and a sharper electronic density of states. This strongly suggests that the isotropic shell growth provides a highly uniform confinement environment for electrons, effectively minimizing electron wave function leakage and energy disorder caused by an irregular shell structure. These findings are fully consistent with the observed prolonged carrier lifetime and spectral narrowing.

Furthermore, in Fig. 3c, the EL spectra of WEC QDs showed a gradual increase in the 400-450 nm region as the driving current was increased, proving that the increase in the number of leakage electrons, which more fully excite the fluorescence emission of TFB. The parasitic emission peak attributed to the recombination of leaked electrons with holes in the HTL eliminated completely in SEC QDs-based devices, also providing compelling evidence that electron leakage has been suppressed by improved electron confinement.

Comment 8: Notably, the use of DPP-Se to control Se reactivity has been reported

multiple times (e.g., Evans et al., *J. Am. Chem. Soc.*, **132**, 10973-10975, 2010), and various alkylamines have also been widely employed (*Front. Chem.*, **6**, 2018, doi:10.3389/fchem.2018.00567). A recent paper (*Nature Communications*, **16**, 1945 (2025)) reported that combining DPP-Se with oleylamine allowed size and shape control of ZnTeSe QDs, yielding improved quality under conditions similar to those here (though n-octylamine was not used). These prior results suggest that the ligand combination in the present study is not entirely unprecedented, though the authors have applied it to produce high-quality materials. A clearer mechanistic explanation of how surface energy was tuned in this work would be beneficial.

Response to comment 8: We sincerely thank the reviewer for this insightful comment. Although the literatures such as *J. Am. Chem. Soc.*, **132**, 10973-10975, 2010, etc. mentioned that DPP-Se can accelerate the reaction kinetics of Se, it cannot overcome the difference in surface energy barriers across crystal facets. As for the recent study (*Nat. Commun.* **16**, 1945 (2025)), although DPP-Se and oleamine were used in the nucleation of both types of ZnSeTe QDs, the IPG QDs presented a cubic shape, while the control QDs showed an irregular shape. The shape control in that work was achieved by introducing a graded-potential shell ($\text{ZnSe}_{0.9}\text{Te}_{0.1}$) to alleviate lattice stress, with no evidence of facet energy modulation being observed. In addition, compared with the TEM images of InP-based QDs synthesized with trioctylamine or oleylamine (*Nature* **575**, 634–638 (2019); *Nature* **635**, 854–859 (2024)), our QDs have a more uniform particle size distribution and higher roundness, thus achieving highly ordered arrangement (**Fig. R12**).

Through binding energy calculation, we confirmed that this is because the binding energy of oleamine and trioctylamine is much lower compared to that of n-octyl amine ligands (**Fig. R13**), which is due to the large steric hindrance hinders their contact with the QD surface, making it ineffective for facet passivation. In contrast, our work precisely utilizes smaller steric hindrance and stronger coordination ability of n-octylamine, synergized with the rapid Se release kinetics of DPP-Se, to achieve kinetic control over shell growth. The synergistic effect of this combination, introduced at the critical early stage of shelling, directly and strongly alters the core's surface energy landscape. This results in QDs with pronounced spherical morphology and a higher degree of ordered assembly, which are critical for achieving superior performance in optoelectronic and display applications. Therefore, our work represents a deliberate and optimized ligand selection strategy that builds upon and advances previous synthetic methodologies.

[FIGURE REDACTED]

Fig. R12. **a**, TEM images of green InP/ZnSe/ZnS QDs (*Nature* **635**, 854–859 (2024)). **b**, STEM images of red InP/ZnSe/ZnS QDs (*Nature* **575**, 634–638 (2019)). **c**, TEM images of InP/ZnSe/ZnS QDs in this work.

Fig. R13. Adsorption model and the binding energy (ΔE) of **a**, tri-octylamine, **b**, oleylamine and **c**, n-octylamine on (111) facet of InP.

Comment 9: In conclusion, while the manuscript presents high-quality materials and strong device results, the novelty of the synthesis approach and the proposed underlying mechanism are not yet fully convincing. I believe the work could be publishable in another journal in its current form, but for *Nature Communications*, additional experimental evidence and mechanistic justification would be necessary to clearly establish the claimed novelty.

Response to comment 9: We sincerely thank the reviewer for their positive assessment of our materials and device performance. We understand and acknowledge your concerns about establishing the mechanistic novelty.

In response to this comment, we have provided a more rigorous theoretical and experimental foundation for the ligand-directed isotropic growth mechanism. New multistage DFT calculations and TEM characterization track the evolution of surface

energies and morphology evolution throughout the synthesis, confirming the critical role of our specific ligand combination (DPP-Se and n-octylamine) in maintaining isotropic growth by minimizing energy differences between different facets. These combined computational and experimental results now provide a comprehensive mechanistic framework that not only explains how our synthesis strategy achieves uniform core-shell structures but also establishes its novelty compared to conventional approaches.

We believe that with this major revision, the manuscript will better meet the high standards of *Nature Communications*.

Response to Reviewer 3:

Comment 1: Guo et al. have synthesized strongly electron-confined green InP QDs towards heavy-metal-free ultrahigh-resolution active-matrix displays. They effectively suppress selective growth of shell on the InP (111) facet through introducing n-octylamine and DPP-Se ligands. High-performing QLEDs based on these QDs exhibit a peak EQE of 23.5%, a high luminance and long operational lifetime. Impressively, the high-resolution QLEDs achieve a compelling 8,460 PPI resolution while maintaining average peak EQEs comparable to thin-film QLEDs. I think this article is interesting and meaningful for the next-generation eco-friendly displays based on heavy-metal-free QDs. I recommend publication after the following issues are addressed.

Response to comment 1: We sincerely thank the Reviewer 3 for the positive evaluation and encouraging comments regarding our work. We are very pleased that the reviewer recognized the innovation and significance of our strategy for synthesizing strongly electron-confined green InP QDs towards heavy-metal-free ultrahigh-resolution active-matrix displays. As noted, the maintaining of high EQE and operational lifetime in microscale patterned QLEDs with a resolution density up to 8,460 PPI demonstrates the potential of this method to address critical challenges in patterned QLEDs and have on advancing impact on next-generation QD displays. Following the Reviewer 3's suggestions, we have carefully revised the manuscript to address the remaining issues, as detailed below.

Comment 2: In this article, the authors mainly focus on the facet-selective growth of shell layers during the synthesis of InP-based QDs. They proposed that the synergistic effect of octylamine and DPP-Se ligands promotes uniform ZnSe shell growth, thereby effectively passivating surface defects. Although it is supported by morphological and spectroscopic data of InP/ZnSe/ZnS QDs, this article still lacks direct evidence for ligand-mediated ZnSe uniformity without additional ZnS shell. It is essential to provide comprehensive characterizations of InP/ZnSe QDs with different ligands (e.g. TEM image, PLQY data, etc.) for the validation of their mechanism.

Response to comment 2: We thank the reviewer 3 for raising this constructive suggestion. To address this, we have supplemented the manuscript with TEM images and PLQY data of InP/ZnSe QDs synthesized with different ligands (**Fig. R14**), thereby providing a more intuitive visualization of the influence of ligands on the uniform growth of the ZnSe shell. The results show that the ligands of n-octylamine can DPP-Se best optimize the morphology of the ZnSe shell, thereby achieving a much higher

PLQY.

Fig. R14. The TEM images and PLQY of InP/ZnSe QDs with different ligands.

To enhance the quality of our manuscript, we have revised relevant discussions in the revised version, as detailed below:

Page 7, Lines 4-7: During the synthesis process, the InP/ZnSe QDs synthesized using the combination of n-octylamine and DPP-Se exhibit higher PLQY (Supplementary Figs. 10 and 11a) and a significantly higher V/D value of 0.59 compared to 0.22 for that synthesized with OA and TOP-Se (Fig. 2a), indicating a more effective and uniform growth of ZnSe.

Comment 3: Moreover, in Supp. Fig. 7, FTIR and XPS data can only demonstrate the existence of octylamine, thus the authors need to add more data to verify the involvement of DPP ligands.

Response to comment 3: We thank the reviewer 3 for raising this constructive suggestion. To characterize the successful coordination of the DPP ligand, we have supplemented the manuscript with Fourier-transform infrared (FTIR) spectroscopy data of WEC and SEC QDs (**Fig. R15**). New characteristic peaks have emerged in the infrared spectra of SEC QDs. The peaks at 750 cm^{-1} and 694 cm^{-1} are characteristic of the C-H bending vibrations of monosubstituted benzene rings, while the bands at 1133 cm^{-1} , 1088 cm^{-1} , and 1043 cm^{-1} correspond to the stretching vibrations of P-Ph bonds.

These characteristic signals provide strong evidence supporting the successful incorporation of DPP.

Fig. R15. Fourier-transform infrared spectra of WEC and SEC QDs.

To enhance the quality of our manuscript, we have revised our manuscript as follows and supplemented related data:

Page 6, Lines 27-28, Page 7, Lines 1-2: Fourier-transform infrared spectra and X-ray photoelectron spectroscopy (XPS) of N 1s and In 3d confirmed that DPP and n-octylamine are successfully introduced and coordinated on the surface of the QDs (Supplementary Fig. 9).

Comment 4: More data or discussions are also needed to further evaluate the effect of octylamine and DPP-Se ligands on controlling facet-selective growth of InP/ZnSe QDs, except XRD and distribution statistics of QD size.

Response to comment 4: To further evaluate the effects of ligands on controlling the facet-selective growth of InP/ZnSe QDs, in addition to XRD and QD size distribution statistics, we defined the degree of isotropic shape through transmission electron microscopy images using a new parameter of circularity (c) (*ACS Energy Letters*, **5**, 1316-1327 (2020)). c is the degree of similarity of the shape to a perfect circle and is calculated as $c = 4\pi(\text{area})/(\text{perimeter})^2$. When the ZnSe shell is grown with ligands of n-octylamine + DPP-Se, the QDs show a uniform and isotropic shape with $c = 0.85$, much higher than other ligands ($c = 0.65$ and $c = 0.77$) (**Fig. R16**). This proves the role of our ligands in promoting the isotropic growth of QDs.

Fig. R16. STEM images and corresponding circularity (c) distributions of the InP/ZnSe/ZnS QDs grown with (a, d) OA + TOP-Se, (b, e) n-octylamine + TOP-Se and (c, f) n-octylamine + DPP-Se.

Comment 5: It is particularly impressive that the SEC QD-based devices demonstrate sustained efficiency exceeding 20% across an extensive pixel size range (1.5-20 μm). For the high performance in high-resolution devices, the authors attributed it to the controlled assembly achieved through their strategy besides the inherent properties of SEC QDs. They need to provide more detailed discussions to clarify the necessity of their assembly strategy.

Response to comment 5: We sincerely thank the reviewer for this positive feedback and for recognizing the significance of our device performance across a wide range of pixel sizes. We also appreciate the valuable suggestion to provide a more detailed discussion elucidating the necessity of our assembly strategy. While the inherent high quality of the SEC QDs is the fundamental basis for high performance, our controlled assembly strategy is indeed the critical enabler for maintaining this high efficiency, especially in sub-20 μm pixels.

For comparison, we also fabricated a QD array via spin-coating. Confocal microscopy images of this control sample (Fig. R17) reveal pronounced coffee-ring effects and inevitable pixel crosstalk, leading to a significant performance roll-off with decreasing pixel size. In contrast, our asymmetric wettability-mediated assembly strategy enables a slower and more controllable solvent evaporation rate, along with directed dewetting

of the three-phase contact line (*Nat Commun.* **16**, 4257 (2025); *Adv. Mater.* **34**, 2110695 (2022)). This effectively suppresses the chaotic fluid flow and coffee-ring formation typically induced by rapid evaporation during spin-coating, resulting in the uniform and dense QD film (Fig. 4b and Supplementary Fig. 27). This uniform QD film improves charge balance and radiation recombination, suppresses electron leakage, and therefore allows for high efficiency even when the pixel size decreases (*Nat Commun.* **16**, 7643 (2025)).

Fig. R17. Fluorescence micrograph of spin-coated micro-LEDs based on SEC QDs.
Scale bar: 20 μm .

To enhance the quality of our manuscript, we have added relevant discussions in the revised version, as detailed below:

Page 12, Lines 11-14: This maintained high performance with decreasing pixel size is attributed to controllable solvent evaporation and directed dewetting in our assembly strategy, which effectively suppresses the pronounced coffee-ring effect and inevitable pixel crosstalk in spin-coating (Supplementary Fig. 30)^{9,19}.

Comment 6: In Supp. Fig. 15, the authors reported the carrier mobility of their core-shell QDs through hole-only/electron-only devices, which exhibit different device structures of QLED device in Fig. 3a. To ensure reproducibility of this article, they need to provide the detailed methodology in both fabrication and measurement of these devices.

Response to comment 6: We sincerely thank Reviewer 5 for raising this critical point. We further elaborated the fabrication and measurement processes of hole-only and electron-only devices in the Experimental section of Supplementary information. The specific revisions are as follows:

Fabrication of HODs. PEDOT:PSS was spin-coated onto ITO substrates at 4000 rpm for 40 s, followed by annealing at 150 °C for 30 min. Then, these substrates were

transferred into a nitrogen-filled glove box for spin-coating. PF8Cz with a concentration of 8 mg mL⁻¹ (solvent: chlorobenzene) was spin-coating at 3,000 rpm for 40 s, and then baked at 120 °C for 30 min. The QD layer was prepared by spin-coating QD solution (20 mg mL⁻¹) at 2,000 rpm for 30 s and baked at 80 °C for 7 min. MoO₃ (with a thickness of 40 nm) and Al electrode (with a thickness of 100 nm) was successively deposited by thermal evaporation under a degree of vacuum of $\approx 2.5 \times 10^{-4}$ Pa. Ultimately, the devices were encapsulated using UV-curable epoxy resin and cover glass in a glove box.

Fabrication of EODs. ZnMgO nanoparticles solution (20 mg mL⁻¹) was spun-coating at 2,000 rpm for 30 s and baked at 60 °C for 30 min. Subsequently, QD layer was prepared by spin-coating QD solution (20 mg mL⁻¹) at 2,000 rpm for 30 s and baked at 80 °C for 7 min, followed by a second spin-coating of the ZnMgO layer in the same parameters as before. Then, Al electrode (with a thickness of 100 nm) was deposited by thermal evaporation under a degree of vacuum of $\approx 2.5 \times 10^{-4}$ Pa. Ultimately, the devices were encapsulated using UV-curable epoxy resin and cover glass in a glove box.

The current density-voltage (*J-V*) curves of the HODs and EODs were measured using a commercial system (XPOY-EQE-Adv, Guangzhou Xipu Optoelectronics Technology Co., Ltd.) by applying DC voltage scans (0 – 8 V, step size 0.2 V). The electron and hole mobilities of both devices were calculated by fitting the space-charge-limited current (SCLC) region, assuming no trap states (see details of calculation in Supplementary Note 6).

Comment 7: Other minor issues should be corrected. For example, the yellow indicators in Fig. 1a are unclear and should be optimized; The color scale in Fig. 2h lacks axis labels; The axis ranges and color bars in Supplementary Figs. 5 and 11 should be unified for better clarity; Unit formats in Supplementary Fig. 11 should be made consistent with the main text; In the figure caption of Supp. Fig. 15, electron-only devices (HODs) is misspelled.

Response to comment 7: We sincerely appreciate Reviewer 3 for his/her careful and thorough inspection. We made point-to-point modifications to these issues as following and correct them in the revised manuscript. We think that these revisions have significantly improved the overall quality of the manuscript.

Fig. R18. Schematic illustration and representative high-resolution TEM images for two types of InP/ZnSe QDs.

To improve our manuscript, we have replaced **Figure 1a** with the new schematic diagram (**Fig. R18**) in the revised manuscript (**Page 19**).

Fig. R19. Temperature-dependent PL spectra of SEC InP/ZnSe/ZnS QDs.

To improve our manuscript, we have replaced **Figure 2h** with the new schematic diagram (**Fig. R19**) in the revised manuscript (**Page 21**).

Fig. R20. Supplementary Fig. 5 | Morphological characterization of QDs synthesized with three combinations of ligands.

Fig. R21. Surface roughness for two types of QD thin films.

To improve our manuscript, we have replaced **Supplementary Fig. 5** and **Supplementary Fig. 11** with the **Fig. R20** and **Fig. R21** in the revised Supplementary Information (**Page 14** and **Page 22**).

Page 28, Lines 4-5: c, d, SCLC measurements of electron-only devices (EODs) based on SEC and WEC QD films.

Response to Reviewers and Revised Details

Response to Reviewer 1:

Comment 1: The authors have thoroughly addressed all the previous concerns and revised the manuscript accordingly. The novelty and significance of the work are now clearly highlighted. I have no further comments, and the manuscript seems suitable for publication in *Nature Communications*.

Response to Comment 1:

We thank the reviewer 1 very much for the positive feedback and their valuable comments throughout the review process!

Response to Reviewer 2:

Comment 1: The authors claim that the combined use of n-octylamine and DPP enables reduced steric hindrance of alkyl chains, improved surface conduction, and ultimately the formation of quantum dots with superior properties compared to those synthesized in a conventional tri-octylamine solvent. They further argue that these improvements are experimentally validated and mechanistically supported through surface-energy calculations on both InP cores and ZnSe shell growth. In particular, the authors' efforts to extend their surface-energy analysis to ZnSe shell formation—and to correlate this with uniform shell growth—are technically sound and represent a commendable attempt to support the proposed mechanism.

Response to Comment 1:

We sincerely appreciate the positive and insightful comments on our manuscript from Reviewer 2. We are especially grateful for the reviewer's recognition of our supplementary experimental verification and mechanical evidence of surface energy modulation and uniform growth of ZnSe shells. The reviewer's affirmation that this approach is "technically sound and represents a commendable attempt" is highly encouraging and validates the significance of this part of our study.

Comment 2: However, despite these strengths, important questions remain regarding the broader impact and originality of the findings. Considering the claimed surface-energy modulation induced by n-octylamine, it is still unclear how the resulting improvements compare with (i) the shell-thickness regimes previously established in the literature, (ii) the commonly reported size distributions of InP/ZnSe(ZnS) QDs, and (iii) the state-of-the-art efficiencies of green InP-based QLEDs. Ligand modifications—whether applied before or after growth—are well-known strategies, and n-octylamine itself is already a reasonable choice within the established ligand library. Thus, it is difficult to identify what fundamentally new concept is being introduced here, especially since the authors' own analysis attributes the strong confinement primarily to uniform shell growth for improved size/shape distribution or any facet-selective growth route.

Response to Comment 2:

We sincerely thank the reviewer 2 for this constructive feedback. To address reviewer's concerns, we compared our SEC QDs with the green InP-based QDs in representative studies in recent years:

(i) Comparison with shell-thickness regimes: Prior literatures have provided useful guidance regarding the relationship between ZnSe shell thickness and electron

confinement. For green InP cores, it has been suggested that a shell thickness of at least 1.47 nm is required to remain 95% probability of electron below the ZnSe shell. Our SEC QDs possess a shell thickness of ~2 nm, well within the range predicted by the referenced study (*ACS Energy Letters*, **5** (4), 1316-1327). In addition, our surface-energy modulation strategy does not seek to replace established shell-thickness control but rather to achieve unprecedented uniformity and conformality of the ZnSe/ZnS shell at any given target thickness. This precise morphological control is critical for consistent charge confinement and transport, directly translating to improved device performance and stability.

(ii) Comparison with size distributions: As shown in the Table R1, the size distribution uniformity of our QDs is the highest among the compared works. This superior morphological control is further corroborated by the TEM images and circularity analysis in our first-round response (Fig. R1). Such enhanced morphological control and ordering are critical for achieving efficient self-assembly and improving the performance of display devices (*Nat Commun* **16**, 4257 (2025); *Nat Commun* **16**, 7643 (2025)).

[FIGURE REDACTED]

Fig. R1. **a**, TEM images of green InP/ZnSe/ZnS QDs (*Nature* **635**, 854–859 (2024)). **b**, STEM images of red InP/ZnSe/ZnS QDs (*Nature* **575**, 634–638 (2019)). **c**, TEM images of InP/ZnSe/ZnS QDs in this work.

(iii) Comparison with state-of-the-art efficiencies: As shown in the Table R1, although the peak EQE (23.5%) reported in our study do not surpass the record-setting values, it still ranks among the highest values reported to date. Crucially, the core originality and broader impact of our work lie in the successful integration of two critical achievements: (i) the synthesis of QDs with superior morphological uniformity via our surface-energy modulation, and (ii) the high device performance can be maintained even under the stringent requirements of pixel-level patterning. The successful application on an active-matrix display provides a scalable template for high-performance, heavy-metal-free, patterned QLED devices.

Table R1. Comparison of size distributions and EQEs in this work with the representative literature on green InP-based QDs.

QD structure	Size distributions	EQE (%)	Year	Ref.
InP/Q-ZnSe/ZnS	5.7 ± 0.8 nm	10.6	2022	1
InP/ZnSe/ZnS	--	15	2022	2
InP/ZnSe _{0.7} S _{0.3} /ZnS	--	15.2	2022	3
InP/ZnSe/ZnS/BDA	8.6 ± 1.2 nm	16.3	2021	4
InP/ZnSe/ZnS	--	6.2	2022	5
InP/ZnSe/ZnS	6.5 ± 0.2 nm	13.8	2023	6
InP/ZnSeS/ZnS	6.31 ± 0.48 nm	5.56	2023	7
InP/ZnSe/ZnS	8.0 ± 1.9 nm	26.68	2024	8
InP/ZnS	--	6.75	2024	9
InP/ZnSe/ZnS	--	12.74	2024	10
InP/ZnSe/ZnSe _x S _{1-x} /ZnS	--	15.4	2025	11
InP/ZnSeS/ZnS	8.0 ± 0.82 nm	26.3	2025	12
InP/ZnSe/ZnS	7.2 ± 0.3 nm	23.5		Our work

References

1. Wu, Q. et al. Quasi-Shell-Growth Strategy Achieves Stable and Efficient Green InP Quantum Dot Light-Emitting Diodes. *Adv. Sci.* **9**, 2200959 (2022).
2. Gao, P. et al. Efficient InP Green Quantum-Dot Light-Emitting Diodes Based on Organic Electron Transport Layer. *Adv. Opt. Mater.* **10**, 2202066 (2022).
3. Yu, P. et al. Highly efficient green InP-based quantum dot light-emitting diodes regulated by inner alloyed shell component. *Light Sci. Appl.* **11**, 162 (2022).
4. Chao, WC. et al. High efficiency green InP quantum dot light-emitting diodes by balancing electron and hole mobility. *Commun. Mater.* **2**, 96 (2021).

5. Sun, Z. et al. Suppressing the Cation Exchange at the Core/Shell Interface of InP Quantum Dots by a Selenium Shielding Layer Enables Efficient Green Light-Emitting Diodes. *ACS Appl. Mater. Interfaces* **14**, 15401–15406 (2022).
6. Wu, Q. et al. Bridging Chloride Anions Enables Efficient and Stable InP Green Quantum-Dot Light-Emitting Diodes. *Adv. Optical Mater.* **11**, 2300659(2023).
7. Zhao, H. One-Pot Synthesis of InP Multishell Quantum Dots for Narrow-Bandwidth Light-Emitting Devices. *ACS Appl. Nano Mater.* **6**, 3797-3802 (2023).
8. Bian, Y. et al. Efficient green InP-based QD-LED by controlling electron injection and leakage. *Nature* **635**, 854-859 (2024).
9. Wang, L. et al. Modified Charge Injection in Green InP Quantum Dot Light-Emitting Diodes Utilizing a Plasma-Enhanced NiO Buffer Layer. *J. Phys. Chem. C* **128**, 3985–3993 (2024).
10. Cheng, Y. et al. High-brightness green InP-based QLEDs enabled by in-situ passivating core surface with zinc myristate. *Mater. Futur.* **3**, 025201 (2024).
11. Luo, Y. et al. Suppression of Interfacial Oxidation in Core/Shell InP Quantum Dots through Solvent Assisted Core-Etching Strategy for Efficient Green Light-Emitting Diodes. *Nano Lett.* **25**, 593-599 (2025).
12. Yuan, C. et al. Dual-Mode Strain Relief via Zinc Acetate Enables High-Efficiency InP Quantum Dot Light-Emitting Diodes. *Angew. Chem. Int. Ed.* **64**, e202509765 (2025).

Comment 3: The authors also argue that the SEC-type QDs yield high performance in unit QD-LED device tests, but such values are comparable to those already reported and do not, on their own, constitute a major conceptual advance suitable for *Nature Communications*. In contrast, the demonstration of good performance in large scale AM-LEDs—indeed unprecedented—could have been a more compelling point of novelty. If the manuscript had focused more deeply on how the SEC QDs maintain their performance during process scaling and pixelation (connection between QDs and large-area/scale LED performances), this could have significantly strengthened its impact.

Response to Comment 3:

We sincerely thank the reviewer for this insightful comment. To better address Reviewer 2's concerns, we analyzing the electroluminescence uniformity of pixels across a range of sizes under confocal microscopy, respectively. The EL mapping and EL intensity curves observed under confocal fluorescence microscopy (**Fig. R2**) show that the green QLEDs of different pixel sizes all exhibit homogeneous emission and

optical flatness, ensuring that the performance does not vary with the reduction of pixels.

This highly uniform assembly is achieved because, compared to conventional deposition methods, our asymmetric wettability-mediated assembly strategy enables a slower and more controllable solvent evaporation rate, along with directed dewetting of the three-phase contact line (*Nat Commun.* **16**, 4257 (2025); *Adv. Mater.* **34**, 2110695 (2022)). This effectively suppresses the chaotic fluid flow and coffee-ring formation typically induced by rapid evaporation during spin-coating, resulting in the uniform and dense QD film (Supplementary Fig. 27). The AFM surface roughness characterization also confirmed this (**Fig. R3**). As the pixel size decreased from 20 μm to 1.5 μm , the pixel surface roughness R_q remained below 1 nm at all times. The uniform spherical morphology of SEC QDs and their well-ordered self-assembly enabled by this controlled assembly process collectively enables uniform electroluminescence across a high-resolution display panel.

Fig. R2. The EL mapping and EL intensity linecut of high-resolution QLEDs with different pixel sizes under confocal microscopy. (a) 20-2 μm , (b) 10-2 μm , (c) 5-2 μm , (d) 3-2 μm , (e) 2-2 μm , (f) 1.5-1.5 μm . The former value represents the side length of

the pixel, and the latter value represents the distance between adjacent pixels. Scale bar, 10 μm .

Fig. R3. Surface roughness of QD arrays for different pixel sizes. AFM image of QD arrays with pixel sizes of (a) 20-2 μm , (b) 10-2 μm , (c) 5-2 μm , (d) 3-2 μm , (e) 2-2 μm , (f) 1.5-1.5 μm . The former value represents the side length of the pixel, and the latter value represents the distance between adjacent pixels. Scale bars, 1 μm (a-c), 200 nm (d and e) and 100 nm (f).

To enhance the quality of our manuscript, we have added relevant discussions in the revised version, as detailed below:

Page 12, Lines 21-23: The uniform spherical morphology of SEC QDs and their well-ordered self-assembly enabled by this controlled assembly process collectively enables uniform electroluminescence across the high-resolution display panel (Supplementary Fig. 31).

Comment 4: Overall, while the manuscript presents scientifically solid work with high-quality materials, strong understanding of the field, and impressive device metrics, the connection between these results and a demonstrably new conceptual advance in AM-LED technology remains insufficiently clear. For these reasons, despite of the appreciation of authors' efforts, I regret to conclude that the work does not meet the novelty and impact threshold required for *Nature Communications*. I recommend considering submission to a more specialized journal where the strengths of the study will be more appropriately recognized.

Response to Comment 4:

We sincerely thank the reviewer for the comprehensive evaluation and for acknowledging that our manuscript presents “scientifically solid work with high-quality materials, strong understanding of the field, and impressive device metrics” We fully understand the concern regarding the connection between our material results and a demonstrably new conceptual advance in AM-LED technology. However, we respectfully wish to clarify that the optimization of quantum dot raw materials and the realization of high-performance AM-LEDs are inextricably linked. The strongly electron-confined InP-based QDs provide a superior material foundation, offering the necessary stability and optoelectronic properties that are prerequisites for high-resolution displays. Additionally, we further demonstrated that our assembly strategy ensures assemble uniformity, enabling the retention of peak EQE and maximum brightness even when pixel sizes are significantly reduced, directly addressing a major challenge in the field where performance typically degrades with scaling.

Based on these comprehensive revisions and the deepened analysis of the underlying mechanisms, we are confident that this work now fully meets the novelty and impact standards required for publication in *Nature Communications*.

Response to Reviewer 3:

Comment 1: The revised manuscript has satisfactorily addressed all of my concerns. It is now suitable for publication in *Nature Communications*.

Response to Comment 1:

We thank the reviewer 3 very much for the positive feedback and their valuable comments throughout the review process!